# One-pot three component synthesis of substituted dihydropyrimidinones using fruit juices as biocatalyst and their biological studies

**Susheel Gulati** [1]*, **Rajvir Singh**[1], **Ram Prakash**[2], **Suman Sangwan**[1]

1 Department of Chemistry, Chaudhary Charan Singh Haryana Agricultural University, Hisar, India,
2 Department of Soil Science, Chaudhary Charan Singh Haryana Agricultural University, Hisar, India

☯ These authors contributed equally to this work.
* sgbhuna108@gmail.com

## Abstract

New and facile one-pot three component approach for the synthesis of substituted dihydropyrimidinones derivatives (4a-4h) from reaction of equimolar substituted aldehydes (1a-1h), methyl acetoacetate (2a) and urea (3a) in presence of nature derived catalyst *viz. Cocos nucifera* L. juice, *Solanum lycopersicum* L. juice and *Citrus limetta* juice, commonly known as coconut juice, tomato juice and musambi juice respectively, at room temperature has been carried out. All synthesized compounds were evaluated for *in vitro* herbicidal activity against *Raphanus sativus* L. (Radish seeds). The compounds (4a-4h) were also screened for their antifungal activity against *Rhizoctonia solani* and *Colletotrichum gloeosporioides* by poisoned food techniques method. Antibacterial activity was also studied against *Erwinia cartovora* and *Xanthomonas citri* by inhibition zone method. From activity data, it was found that compounds 4g and 4d were most active against *Raphanus sativus* L. (root) and *Raphanus sativus* L. (shoot) respectively. Compounds 4f and 4c was found most active against *Rhizoctonia solani* and *Colletotrichum gloeosporioides* fungus respectively at highest concentration. Compound 4g has shown maximum inhibition zone i.e. 1.00–5.50 mm against *Erwinia cartovora* at 2000 μg/mL concentration. Maximum *Xanthomonas citrii* growth was inhibited by compounds 4f showing inhibition zone 4.00–12.00 mm at highest concentration. Short reaction time, high yields, mild reaction condition and simple work-up are some merits of present methodology.

## Introduction

Dihyropyrimidinones and its derivatives are found in a large family of natural products with wide biological activities, due to which they become important class of heterocyclic compounds. They generally possess therapeutic and pharmacological properties [1] and several derivatives of dihydropyrimidinone acts as calcium channel modulators [2], Ca-antagonists and vasodilative and antihypertensive [3]. In 1893, Biginelli reported the synthesis of

**Data Availability Statement:** All relevant data are within the paper and its supporting information files.

**Funding:** Authors received no specific funding for this study.

**Competing interests:** The authors have declared that no competing interests exist.

dihyropyrimidinones, which is one-pot three component reaction between an substituted aldehydes, β-ketoester and urea. The major drawbacks of this procedure are lower yields and longer reaction time [4]. Therefore, we need to perform a very mild, simple, cost-effective, commercially beneficial and eco-friendly procedure for synthesis of dihydropyrimidinone derivatives for the academia and pharmaceutical industries [5]. Recently an efficient and facile green method has been developed for synthesizing substituted dihydropyrimidinones *via* Biginelli reaction at room temperature in presence of fruit juices *viz* amla, orange and lime juice [6]. Fruit juice is also acts as natural acids which were used as biocatalyst in organic synthesis. Now a day's fruit juice used in organic synthesis as homogeneous catalysts for various selective transformations of simple and complex molecules. The growing interest in fruit juice is mainly because of its biocatalysts, eco-friendly nature, non-toxic, non-hazardous and cost effective [7]. Therefore, in this paper we reported one-pot three components clean and facile synthesis of dihydropyrimidinone derivative in presence of fruit juices *viz. Cocos nucifera* L., *Solanum lycopersicum* L., *Citrus limetta* respectively. Some biologically and pharmacologically active dihydropyrimidinones derivatives are shown in Fig 1.

## Materials and methods

All the chemicals were purchased from CDH (Central Drug House), SRL (Sisco Research Laboratory) and Sigma-Aldrich. Reaction were performed in oven-dried glassware and monitored by thin layer chromatography (TLC) silica gel plates using ethylacetate in hexane and other solvents combinations as the mobile phase. Melting points were determined in open capillaries on a Ganson electric melting point apparatus and are uncorrected. The $^1$HNMR spectra were recorded in CDCl$_3$ or DMSO-$d_6$ using tetra methyl silane (TMS) as internal reference on "Brucker Ac 400 F"(400MHz) nuclear magnetic resonance spectrometer. The chemical shifts values are quoted in delta (parts per million, ppm). Infrared spectra (4000–350 cm$^{-1}$) of the synthesized compounds were recorded in KBr pellets on Perkin Elmer FTIR-R2X spectrophotometer and frequency is expressed in cm$^{-1}$.

### Biological studies

**Herbicidal activity.**   Solutions of 50 μg/ mL, 100 μg/ mL, 150 μg/ mL and 200 μg/ mL of the test compounds in DMSO were prepared. Agar powder (5g) was put into boiling distilled water (1L) until it dissolved, and then cooled down to 40–50˚C. The solution (2mL) containing test compounds and melting agar (18mL) was mixed and this mixture was added to a Petridish with 4.5 cm diameter. The agar plate without test compound was used as an untreated control. Then 15 seeds of *Raphanus Sativus* L. (Radish) were put on the surface of the agar plate. The Petridishes were covered with glass lids, and the cultivation conditions were kept at 25±1˚C and 12 hours in light and 12 hours in dark alternating for seven days. Seven days later, the root lengths and shoot lengths of *Raphanus sativus* L. were measured. The growth inhibitory rate related to untreated control was determined by given formula [8].

$$\%\text{Inhibition} = \frac{\text{Control} - \text{Treated}}{\text{Control}} \times 100$$

**Antifungal activity.**   All synthesized compounds (4a-4h) were tested for their antifungal activity against *Rhizoctonia solani* and *Colletotrichum gloeosporioides* respectively. Fungal species were grown in laboratory on Potato dextrose agar (PDA) media. The antifungal activity was determined by poisoned food technique method [9]. The required amount of synthesized compounds dissolved in 1 mL of DMSO was incorporated aseptically into 99 mL aliquots of

Fig 1. Some medicinally important substituted dihyropyrimidinones.

sterilized potato dextrose agar cooled at 45˚C after brief shaking. Each lot of medium was poured into Petri dishes and allowed to solidify. DMSO was used as negative control. Each dish was inoculated centrally with a 5 mm mycelial disc cut from the periphery of 2–3 days old fungal colonies. Inoculated Petri plates were incubated in the dark 25±2˚C for 48–72 h and colony diameters were measured periodically till the control dishes were nearly completely covered with fungus growth. All observations were made in triplicate. The degree of inhibition of growth was calculated from the mean differences between treatments and the control as percentage of latter by using the formula.

$$\%\text{Inhibition} = \frac{\text{Control} - \text{Treated}}{\text{Control}} \times 100$$

Control = mycelial growth in control dish

Treated = mycelial growth in treated dish

**Antibacterial activity.**   Bacterial species *Erwinia cartovora* and *Xanthomonas citri* were grown on Luria-Bertani medium in laboratory. Antibacterial activity was evaluating using inhibition zone method [10]. 250, 500, 1000 and 2000 μg/ mL concentrations of synthesized compounds were prepared from the stock solution by taking appropriate amount and diluting with DMSO. DMSO was used as negative control. The circular paper discs of 10 mm diameter were prepared from Whatman's Filter paper No. 1. The disc were kept in Petri plate and autoclaved at 15 lbs pressure 20 minutes. Two paper discs were used for each concentration of the synthesized compounds. The excess of solution absorbed by paper discs was removed by holding them vertically by sterile forecep. Such soaked discs were transferred aseptically to Petri

plates containing media and bacterial suspension spread over the surface. The Petri plates were kept in an incubator at 25±2°C overnight and then examined for inhibition zone at these different concentrations of compounds. The experiment was performed in triplicate and activity was determined on the basis of inhibition zone (in mm).

**Statistical analysis.** The experiments were performed in triplicates for each treatment and the mean value were recorded and expressed as mean ± S.D. The descriptive statistics in form of box-and-whisker diagram were also presented in this paper. The spacing between the different parts of the box indicates the degree of dispersion and skewness in the data.

**Preparation of biocatalyst.** *Extraction of Cocos nucifera L. juice.* The coconut juice was obtained by perforating the fruit with a knife. The coconut juice was filtered using filter paper whatman no 1 for the elimination of residues to get clear juice which was used as a catalyst [11].

*Preparation of Solanum lycopersicum L. juice.* Fresh tomatoes were purchased from the local market. Then washed thoroughly under running tap water followed by rinsing thrice with double distilled water. Tomatoes were squeezed and juice were strained initially through a muslin cloth then passed through Whatman filter paper No. 1 [12].

*Preparation of Citrus limetta juice.* First of all wash the sweet limes and pat them dry. Cut them into two halves. Then using a citrus juice squeezer, extract the juice. Then the juice was filtered through cotton and then through whatman filter paper no 1 to remove solid material and to get clear juice which as used as a catalyst [13].

**General procedure for the synthesis of substituted dihydropyrimidinones derivatives (4a-4h).** *By Cocos nucifera L. juice (Method A).* The mixture of 20 mmol of substituted aldehydes (1a-1h), 20 mmol of methyl acetoacetate (2a), 20 mmol of urea (3a) and 2.5 mL of *Cocos nucifera* L. juice was stirring at room temperature. The completion of reaction was monitored by TLC. Then the reaction mixture was filtered, washed with water and pure yellow crystalline solid (4a-4h) recovered by recrystallized with methanol. All synthesized compounds were confirmed by FTIR and NMR and its melting point.

*By Solanum lycopersicum L. juice (Method B).* A mixture of substituted aldehydes (20 mmol) (1a-1h), methyl acetoacetate (20 mmol) (2a), urea (20 mmol) (3a) and *Solanum lycopersicum* L. juice (10 mL) was taken in round-bottom flask and stirring at room temperature. The progress of reaction was monitored using thin layer chromatography. Then the reaction mixture was filtered, washed with water and pure yellow crystalline solid (4a-4h) recovered by recrystallized with methanol. All synthesized compounds were confirmed by FTIR and NMR and its melting point.

*By Citrus limetta juice (Method C).* In 50 mL round bottom flask, substituted aldehydes (20 mmol) (1a-1h), methyl acetoacetate (20 mmol) (2a), urea (20 mmol) (3a) and *Citrus limetta* juice (10 mL) was stirred till the completion of reaction as indicated by thin layer chromatography. Then the reaction mixture was filtered, washed with water and pure yellow crystalline solid (4a-4h) recovered by recrystallized with methanol. All synthesized compounds were confirmed by FTIR and NMR and its melting point.

All the dihydropyrimidinone derivatives (4a-4h) were prepared according to Method A, B and C.

## Results and discussion

We reported the synthesis of substituted dihydropyrimidinones (4a-4h) by one-pot multicomponent reaction between equimolar substituted aldehydes (1a-1h) *viz*. 2-Hydroxybenzaldehyde (1a), 4-Methoxybenzaldehyde (1b), 3,4-Dimethoxybenzaldehyde (1c), 4-Chlorobenzaldehyde (1d), 4-Bromobenzaldehyde (1e), 3-Hydroxybenzaldehyde (1f),

**Scheme 1. Synthesis of substituted dihyropyrimidinones (4a-4h).**

2-Chlorobenzaldehyde (1g) and 4-Methylbenzaldehyde (1h) methyl acetoacetate (2a) and urea (3a) in presence of green catalyst *viz. Cocos nucifera* L. juice, *Solanum lycopersicum* L. juice and *Citrus limetta* juice at room temperature (Scheme 1).

The reaction conditions were optimized by performing the reaction between 4-Hydroxy-3-methoxybenzaldehyde (20 mmol), methyl acetoacetate (20 mmol) and urea (20 mmol) in presence of catalyst *viz. Cocos nucifera* L. juice, *Solanum lycopersicum* L. juice and *Citrus limetta* juice at room temperature. Initially, the reaction was attempted between 4-Hydroxy-3-methoxybenzaldehyde (20 mmol), methyl acetoacetate (20 mmol) and urea (20 mmol) in presence of *Cocos nucifera* L. juice at room temperature. It was found that excellent yield (86%) of product was obtained when amount of *Cocos nucifera* L. juice was 2.5 mL and reaction time was also reduced (Table 1, Entry 4). These results indicated that 2.5 mL of *Cocos nucifera* L. juice gives high yield of product over the completion of the reaction. Further the reaction was also optimized in presence of *Solanum lycopersicum* L. juice and *Citrus limetta* juice respectively. When the concentration of *Solanum lycopersicum* L. juice in the reaction mixture was 10.0 mL than yield of reaction was maximum (91%) and reaction completed in 5h (Table 2, Entry 4). Hence 10.0 mL of *Solanum lycopersicum* L. juice under aqueous conditions at room temperature is the optimal condition for this reaction. The same reaction was also screened under aqueous conditions at room temperature in presence of *Citrus limetta* juice and maximum yield 93% (Table 2, Entry 4) was obtained when the amount of *Citrus limetta* juice was 10.0 mL in reaction mixture. The physical data of this study are presented in Table 3. After completion of the reaction, the solid products was collected by simple filtration and then recrystallized in methanol to afford pure dihydropyrimidinone derivatives (4a-4h). The structure of synthesized compounds was confirmed by [1]HNMR, FTIR analysis as well as

**Table 1. Model reaction using *Cocos nucifera* L. juice as catalyst.**

| Entry | Catalyst Concentration (mL) | Method A | |
|---|---|---|---|
| | | Time (min) | Yield (%) |
| 1 | 1.0 | 60 | 69 |
| 2 | 1.5 | 30 | 71 |
| 3 | 2.0 | 20 | 75 |
| **4** | **2.5** | **10** | **86** |

**Table 2. Model reaction using *Solanum lycopersicum* L. juice and *Citrus limetta* juice as catalyst.**

| Entry | Catalyst Concentration (mL) | Method B | | Method C | |
|---|---|---|---|---|---|
| | | Time (h) | Yield (%) | Time (h) | Yield (%) |
| 1 | 4.0 | 14 | 11 | 12 | 62 |
| 2 | 6.0 | 12 | 76 | 9 | 85 |
| 3 | 8.0 | 9 | 90 | 6 | 91 |
| **4** | **10.0** | **5** | **91** | **4.5** | **93** |

comparison of their melting points with those of reported compounds. The comparison of activity of different catalysts with respect to time and yield of reaction as shown in Table 4. All synthesized substituted dihydropyrimidinones derivatives (4a-4h) were shown in Fig 2. From spectral study it was found that compound *viz*. Methyl 4-(2-hydroxyphenyl)-6-methyl-2-oxo-1,2,3,4-tetrahydropyrimidine-5-carboxylate (4a) showed $^1$H NMR spectrum in DMSO-$d_6$, displayed a singlet at 1.75 δ integrating for three proton of methyl group, singlet at 3.71 δ integrating for three proton of COOCH$_3$ group, singlet at 4.51 δ integrating for one proton of OH group, multiplet at 6.73–7.19 δ integrating for proton of aryl group, singlet at 7.46 δ integrating for one proton of NH group, singlet at 7.69 δ integrating for one proton of NH group and melting point 198–200˚C. The compound *viz*. Methyl 6-(4-methoxyphenyl)-4-methyl-2-oxo-1, 2-dihydropyrimidine-5-carboxylate (4b) showed $^1$H NMR spectrum in DMSO-$d_6$, displayed a singlet at 2.25 δ integrating for three proton of methyl group, singlet at 3.72 δ integrating for three proton of aryl methoxy group, singlet at 3.51 δ integrating for three proton of COOCH$_3$ group, multiplet at 6.57–7.66 δ integrating for proton of aryl group, singlet at 5.11 δ integrating for one proton of NH group, singlet at 9.16 δ integrating for one proton of NH group and melting point 204–205˚C. The compound *viz*. Methyl 4-(3,4-dimethoxyphenyl)-6-methyl-2-oxo-1,2,3,4-tetrahydropyrimidine-5-carboxylate (4c) showed $^1$H NMR spectrum in DMSO-$d_6$, displayed a singlet at 2.51 δ integrating for three proton of methyl group, singlet at 3.82 δ integrating for three proton of aryl methoxy group, singlet at 3.86 δ integrating for three proton of aryl methoxy group, multiplet at 7.13–7.55 δ integrating for proton of aryl group, singlet at 5.48 δ integrating for one proton of NH group, singlet at 9.83 δ integrating for one proton of NH group and melting point 198˚C. The compound *viz*. Methyl 6-(4-chlorophenyl)-4-methyl-2-oxo-1,2-dihydropyrimidine-5-carboxylate (4d) displayed IR absorptions at 3415.3, 3325.7, 1680.5, 1591.5 and 761.0 cm$^{-1}$ indicating the presence of NH, NH, C = O, C = C aromatic and C-Cl respectively. The compound *viz*. Methyl 6-(4-bromophenyl)-4-methyl-2-oxo-1,2-dihydropyrimidine-5-carboxylate (4e) displayed IR absorptions at 3436.6, 3307.5, 1696.2, 1587.8

**Table 3. Physical data of substituted dihyropyrimidinones (4a-4h).**

| Entry | Product | R | R$^1$ | R$^2$ | Method A | | Method B | | Method C | | mp ºC lit mp$^{ref}$ |
|---|---|---|---|---|---|---|---|---|---|---|---|
| | | | | | Time (min) | Yield (%) | Time (h) | Yield (%) | Time (h) | Yield (%) | |
| 1 | **[4b]** | [2-OHPh] | OCH$_3$ | CH$_3$ | 10 | 81 | 8 | 86 | 5 | 78 | 198–200, 200–202 [15] |
| 2 | **[4a]** | [4-OCH$_3$Ph] | OCH$_3$ | CH$_3$ | 10 | 82 | 8 | 87 | 8 | 81 | 204–205, 201–203 [16] |
| 3 | **[4c]** | [3,4-OCH$_3$Ph] | OCH$_3$ | CH$_3$ | 10 | 84 | 6 | 80 | 4 | 82 | 198 |
| 4 | **[4d]** | [4-ClPh] | OCH$_3$ | CH$_3$ | 10 | 90 | 2 | 86 | 4 | 75 | 180–182, 182–184 [17] |
| 5 | **[4e]** | [4-BrPh] | OCH$_3$ | CH$_3$ | 10 | 85 | 2 | 88 | 4 | 80 | 205 |
| 6 | **[4f]** | [3-OHPh] | OCH$_3$ | CH$_3$ | 10 | 78 | 6 | 90 | 5 | 87 | 201 |
| 7 | **[4g]** | [2-ClPh] | OCH$_3$ | CH$_3$ | 30 | 80 | 4 | 83 | 5 | 92 | 202–204, 205–206 [18] |
| 8 | **[4h]** | [4-CH$_3$Ph] | OCH$_3$ | CH$_3$ | 60 | 87 | 4 | 80 | 4 | 81 | 208 |

**Table 4. Comparison for different catalysts used for synthesis of dihyropyrimidinones (4a-4h).**

| S.No | Catalyst | Reaction condition | Time (h) | Yield (%) | References |
|------|----------|-------------------|----------|-----------|------------|
| 1 | $p$-TSA | Refluxed in EtOH | 1.0 | 90 | [19] |
| 2 | ZnCl$_2$ | MW Irradiation | 30 sec | 94 | [20] |
| 3 | Zn(BF)$_4$ | Stirring at RT | 4.0 | 71 | [21] |
| 4 | Y(OAC)$_3$ | 115°C | 4.5 | 89 | [22] |
| 5 | Mg(NO$_3$)$_2$ | Refluxed | 45 min | 90 | [23] |
| 6 | CaCl$_2$ | Refluxed in EtOH | 2.0 | 98 | [24] |
| 7 | InBr$_3$ | Refluxed in EtOH | 7.0 | 97 | [25] |
| 8 | Pb(NO$_3$)$_2$ | Refluxed in CH$_3$CN | 3.0 | 89 | [26] |
| 9 | P$_2$O$_5$ | Refluxed at 100°C | 1.5 | 94 | [27] |
| 10 | Citric acid | 80°C | 1.0 | 79 | [28] |
| **11** | ***Cocos nucifera* L. juice** | **RT** | **10 min** | **86** | **Present work** |
| **12** | ***Solanum lycopersicum* L. juice** | **RT** | **5.0** | **91** | **Present work** |
| **13** | ***Citrus limetta* juice** | **RT** | **4.5** | **93** | **Present work** |

and 811.7 cm$^{-1}$ indicating the presence of NH, NH, C = O, C = C aromatic and C-Br respectively and melting point 205°C. The compound *viz.* Methyl6-(3-hydroxyphenyl)-4-methyl-2-oxo-1,2-dihydropyrimidine-5-carboxylate (4f) displayed IR absorptions at 3227.7, 3106.4, 3372.4, 1714.0 and 1487.9 cm$^{-1}$ indicating the presence of NH, NH, OH, C = O and C = C

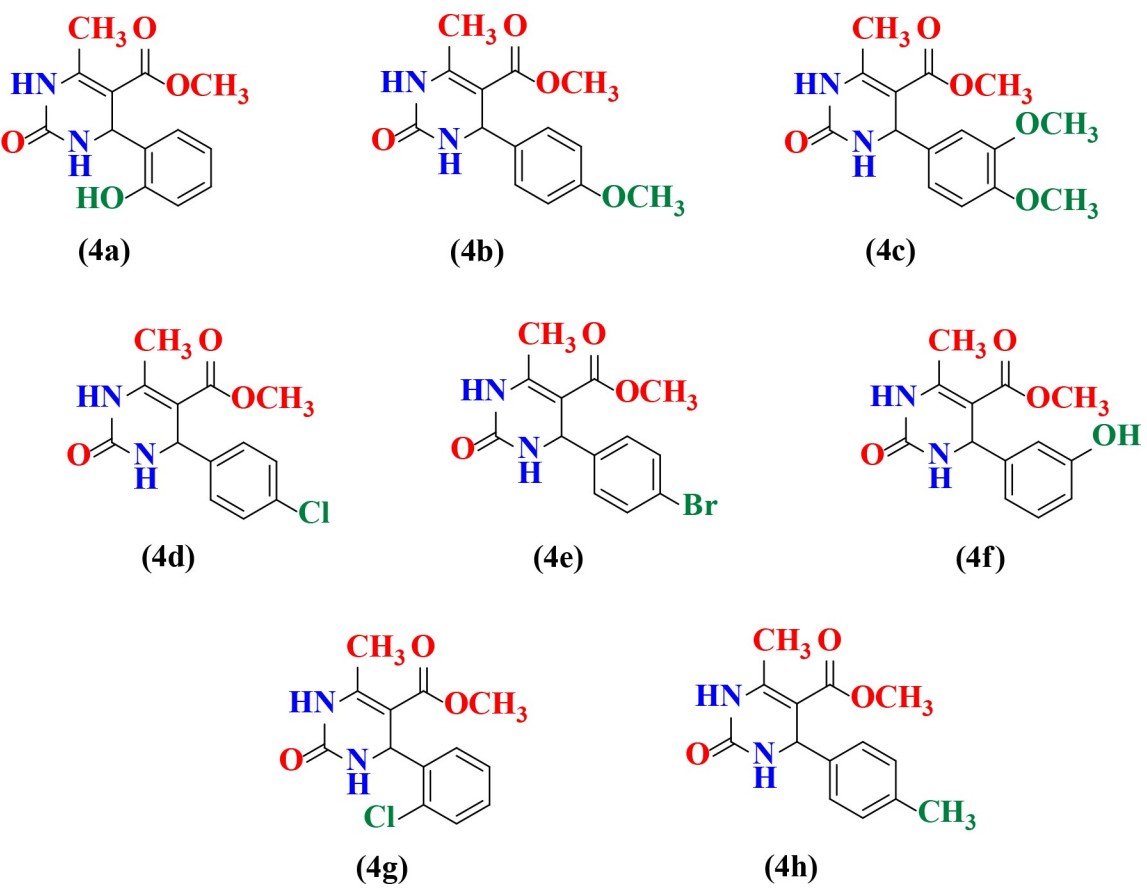

**Fig 2. Substituted dihydropyrimidinones derivatives (4a-4h).**

aromatic respectively and melting point 201˚C. The compound *viz*. Methyl 6-(2-chlorophe-nyl)-4-methyl-2-oxo-1,2-dihydropyrimidine-5-carboxylate (4g) showed [1]H NMR spectrum in DMSO-$d_6$, displayed a singlet at 2.31 δ integrating for three proton of methyl group, singlet at 3.46 δ integrating for three proton of COOCH$_3$ group, multiplet at 7.20–7.36 δ integrating for proton of aryl group, singlet at 5.65 δ integrating for one proton of NH group, singlet at 9.27 δ integrating for one proton of NH group and melting point 202–204˚C. The compound *viz*. Methyl 4-methyl-2-oxo-6-(p-tolyl)-1,2-dihydropyrimidine-5-carboxylate (4h) displayed IR absorptions at 3484.4, 3335.7, 1740.5 and 1651.7 cm$^{-1}$ indicating the presence of NH, NH, C = O and C = C aromatic respectively and melting point 208˚C. We found that *Cocos nucifera* L. juice, *Solanum lycopersicum* L. juice and *Citrus limetta* juice gives the best catalytic activity in terms of product yield, reaction condition and reaction time compared to other catalyst in literature *viz. p*-TSA, ZnCl$_2$, Zn(BF)$_4$, Y(OAC)$_3$, Mg(NO$_3$)$_2$, CaCl$_2$, InBr$_3$, Pb(NO$_3$)$_2$, P$_2$O$_5$ and Citric acid. The catalyst used in present study is nature derived, easily available and cost effective which makes this procedure is eco-friendly. In summary, we have reported a facile one-pot three component synthesis of substituted dihydropyrimidinones derivatives by condensation of substituted aldehyde, methyl acetoacetate and urea at room temperature in presence of *Cocos nucifera* L. juice, *Solanum lycopersicum* L. juice and *Citrus limetta* juice in excellent yields. The current procedure offers many advantages such as simple and efficient catalytic system, simple work-up, no use of hazardous solvents, cheap and products are obtained in good to excellent yields. Moreover, all products were obtained through simple filtration with no need for column chromatography, which reduces the waste as well as environmental pollution. We also conclude that current protocol will provide great utility in the synthesis of other heterocyclic compounds in the near future. The proposed mechanism for the formation of substituted dihydropyrimidinones is shown in Scheme 2.

The first mechanism *via* the iminium route involves condensation between aldehyde and urea to form iminium intermediate (D), which undergoes nucleophillic addition with a β-keto ester leading to DHPMs. The second mechanism *via* enamine route (E) involves condensation reaction between urea and β-keto ester form protonated enamine intermediate, which reacts with aldehyde to give DHPMs. The third mechanism involves Knoevenagel condensation reaction (F) between aldehyde and β-keto ester results in the formation of carbenium ion intermediate, which further react with urea to give DHPMs [14]. It was also found that iminium mechanism is the most favourable route, both kinetically and thermodynamically to form DHPMs derivatives. Thus both experimental and theoretical results support the mechanism *via* iminium route as most plausible one for Biginelli reaction.

## Herbicidal activity

All compounds (4a-4h) were tested for herbicidal activity against *Raphanus sativus* L. at 200, 150, 100 and 50 μg/mL concentrations as shown in Table 5. Results were shown in the form of primary screening. All compounds were diluted to 1000 μg/mL concentration as a stock solution. Herbicidal activities of compounds were evaluated against *Raphanus sativus* L. by inhibitory effect of compounds on the growth of weed roots and shoots. The percentage of inhibition growth was calculated from mean differences between treated and control. From the herbicidal activity results, we observed that compound 4g was exhibited maximum percentage growth inhibition i.e. 93.33 against *Raphanus sativus* L. (root) whereas compound 4d was exhibited maximum percentage growth inhibition i.e. 87.50 against *Raphanus sativus* L. (shoot) respectively at 200 μg/mL concentrations. The compounds 4g and 4d showed broadspectrum herbicidal activity because of presence of chloro substitution at phenyl ring. The herbicidal activity of compounds is given in Fig 3. The box plot and graphical representation of herbicidal activity of all compounds against *Raphanus sativus* L. were shown in Figs 4–7.

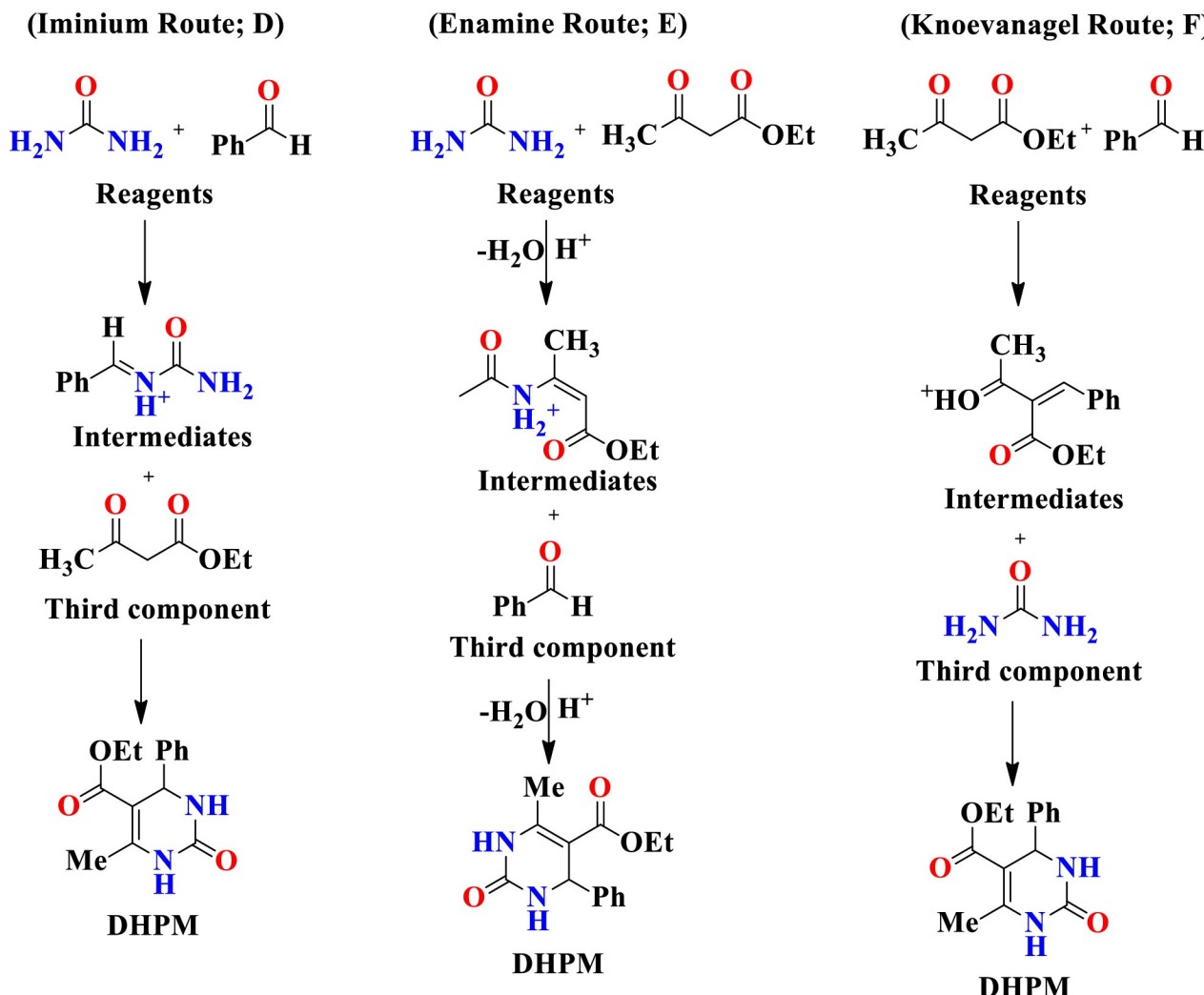

**Scheme 2. Proposed mechanism for formation of substituted dihyropyrimidinones (4a-4h).**

**Table 5. Herbicidal activity of substituted dihydropyrimidinones (4a-4h).**

| Compounds | Growth Inhibition (%) | | | | | | | |
|---|---|---|---|---|---|---|---|---|
| | Root | | | | Shoot | | | |
| | 50 (µg/mL) | 100 (µg/mL) | 150 (µg/mL) | 200 (µg/mL) | 50 (µg/mL) | 100 (µg/mL) | 150 (µg/mL) | 200 (µg/mL) |
| 4a | 25.00 ± 0.86 | 50.00 ± 0.83 | 66.60 ± 1.05 | 83.33 ± 1.26 | 47.50 ± 0.46 | 56.25 ± 0.85 | 71.25 ± 1.00 | 81.25 ± 1.07 |
| 4b | 33.33 ± 1.00 | 50.00 ± 2.00 | 75.00 ± 0.99 | 91.66 ± 1.07 | 56.25 ± 0.72 | 68.75 ± 0.74 | 81.20 ± 1.01 | 86.25 ± 0.56 |
| 4c | 30.20 ± 1.02 | 49.36 ± 1.09 | 74.89 ± 1.24 | 92.36 ± 0.99 | 54.27 ± 0.90 | 68.12 ± 1.15 | 78.78 ± 0.58 | 87.36 ± 0.93 |
| 4d | 16.60 ± 1.00 | 33.33 ± 0.94 | 58.33 ± 1.34 | 83.33 ± 0.07 | 37.50 ± 0.99 | 52.50 ± 1.21 | 68.75 ± 0.74 | 87.50 ± 1.07 |
| 4e | 18.67 ± 0.52 | 37.38 ± 0.80 | 59.64 ± 1.27 | 87.52 ± 0.53 | 16.66 ± 1.80 | 35.48 ± 0.68 | 56.97 ± 0.76 | 81.87 ± 1.08 |
| 4f | 38.72 ± 0.90 | 53.16 ± 0.99 | 70.48 ± 1.00 | 89.38 ± 0.93 | 36.78 ± 0.67 | 48.18 ± 0.85 | 66.66 ± 0.74 | 84.78 ± 1.15 |
| 4g | 66.60 ± 0.53 | 76.60 ± 1.00 | 86.60 ± 0.47 | 93.33 ± 1.46 | 47.69 ± 0.78 | 61.50 ± 0.85 | 72.30 ± 1.07 | 84.61 ± 1.06 |
| 4h | 50.00 ± 0.82 | 66.60 ± 1.12 | 80.00 ± 0.84 | 90.00 ± 1.26 | 38.46 ± 1.49 | 53.80 ± 1.00 | 66.15 ± 0.86 | 83.07 ± 1.71 |

All values are mean ± S.D.

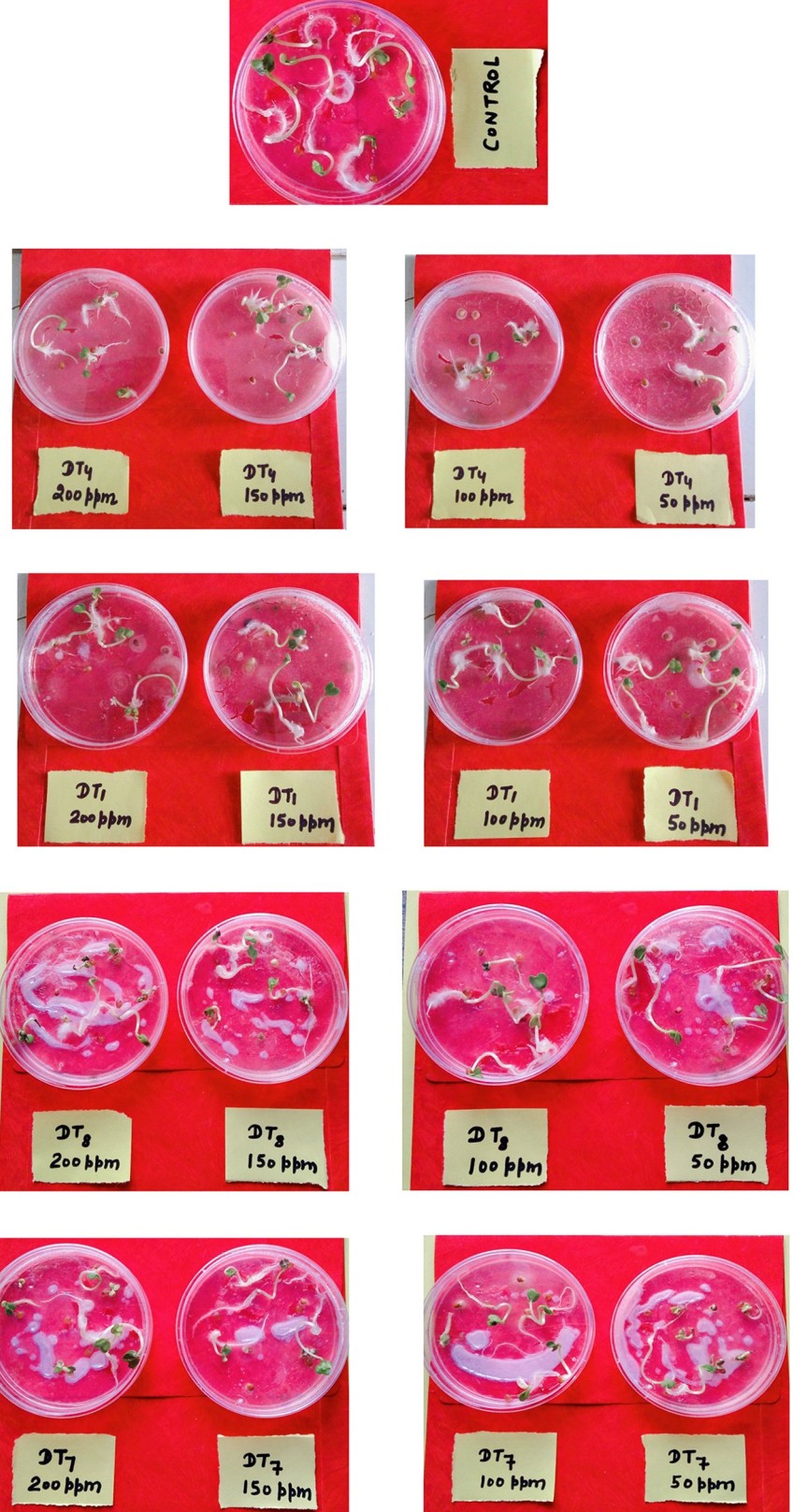

**Fig 3. Herbicidal activity of substituted dihydropyrimidinones (4a-4h).**

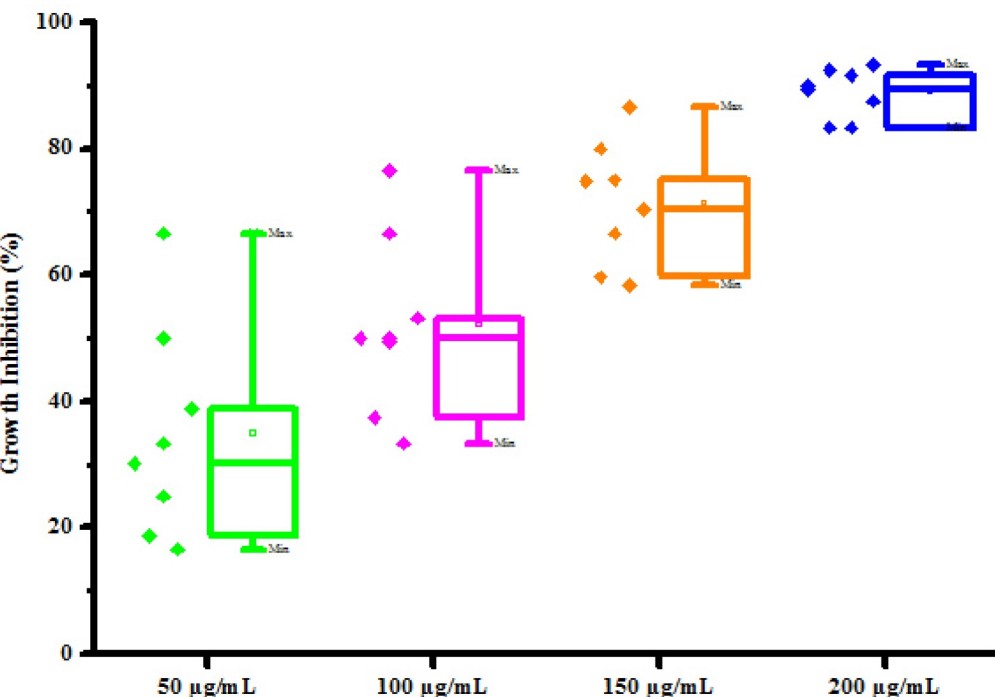

**Fig 4. Box plot of substituted dihyropyrimidinones (4a-4h) against Raphanus sativus L. (root).**

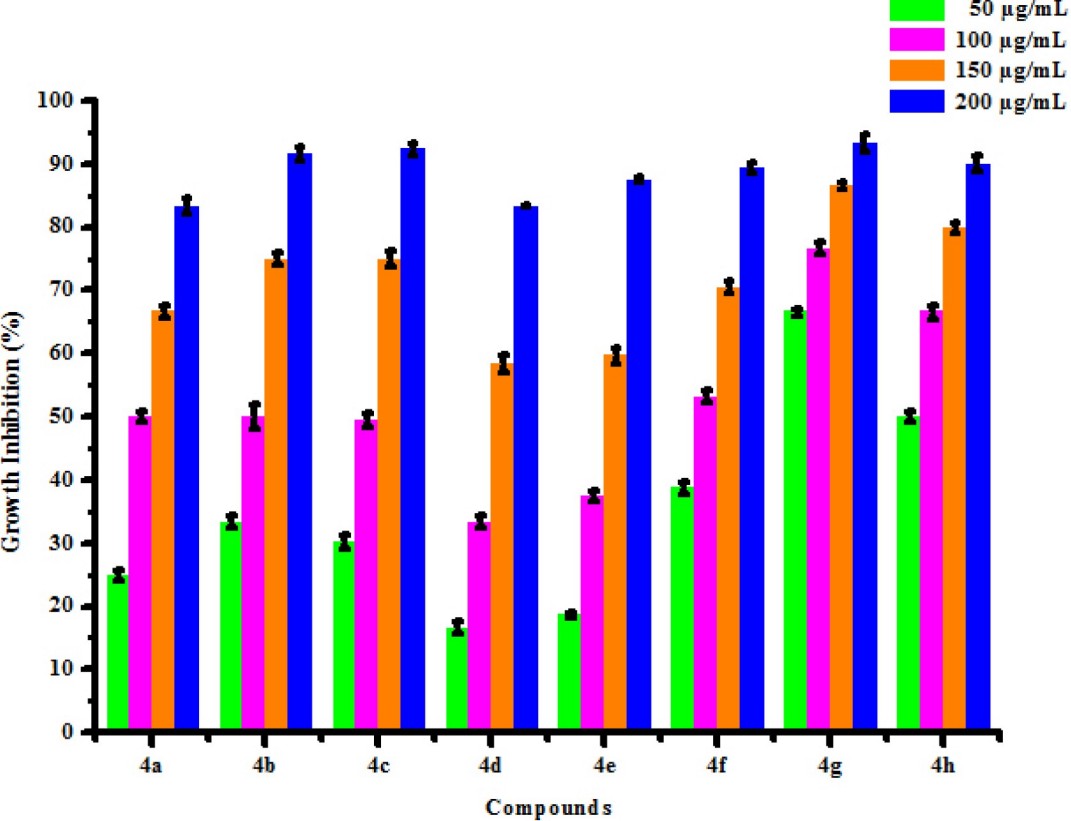

**Fig 5. Herbicidal activity of substituted dihyropyrimidinones (4a-4h) against *Raphanus sativus* L. (root).**

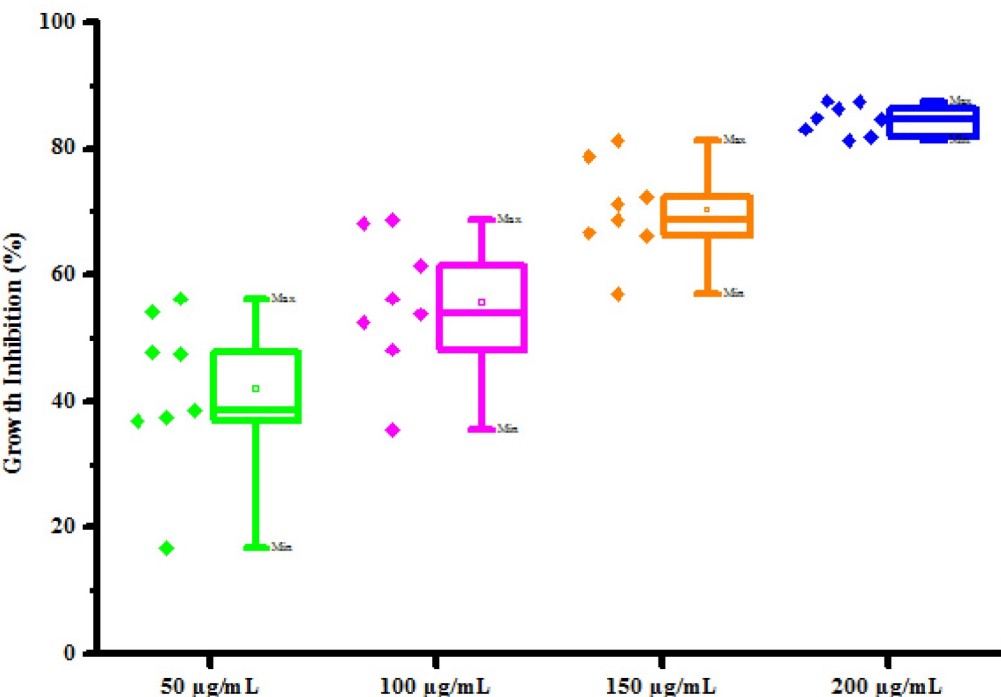

**Fig 6. Box plot of substituted dihyropyrimidinones (4a-4h) against *Raphanus sativus* L. (shoot).**

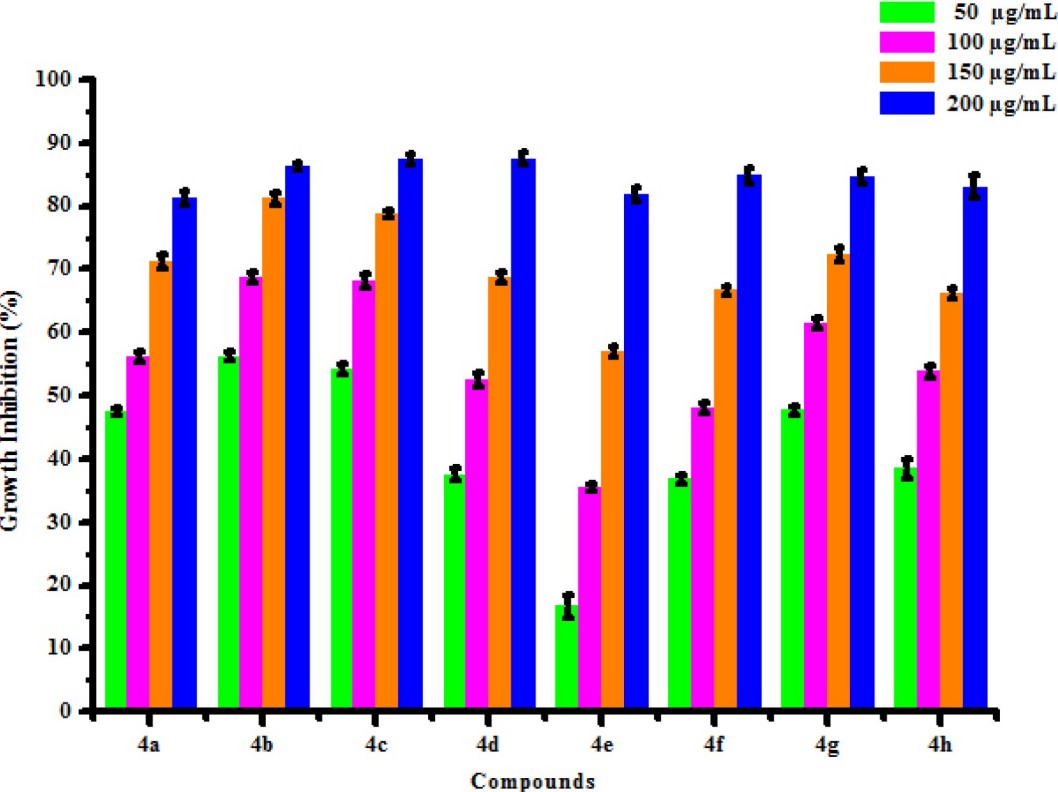

**Fig 7. Herbicidal activity of substituted dihyropyrimidinones (4a-4h) against *Raphanus sativus* L. (shoot).**

## Antimicrobial evaluation

### Antifungal activity

All synthesized compounds (4a-4h) were screened for their antifungal activity against 2 fungal strains *viz. Rhizoctonia solani* and *Colletotrichum gloeosporioides* by poisoned food technique method. DMSO was used as negative control against fungal strains. The result of antibacterial activity of tested compounds is shown in Table 6. Most of synthesized compounds possess a moderate to good activity against *R. solani* and *C. gloeosporioides* respectively. Compounds (4a) and (4b) was found active against *Rhizoctonia solani* fungus at 250, 500, 1000 and 2000 μg/mL concentrations showing percentage growth inhibition 61.53, 71.15, 80.76, 86.53% and 41.35, 62.56, 78.13, 89.38% respectively. Compounds (4c), (4d), (4e) and (4f), (4g), (4h) has been also shown growth inhibition 50.87, 69.99, 80.00, 91.13%, 40.38, 65.38, 80.76, 90.38%, 35.70, 54.68, 67.30, 79.89% and 55.50, 70.50, 80.98, 92.00%, 48.27, 65.51, 75.86, 82.75%, 60.78, 70.58, 82.35, 90.19% respectively against *Rhizoctonia solani* at different concentrations 250, 500, 1000 and 2000 μg/mL. Compounds (4a) and (4b) was found active against *Colletotrichum gloeosporioides* fungus at 250, 500, 1000 and 2000 μg/mL concentrations showing percentage growth inhibition 37.89, 53.80, 78.18, 89.78% and 35.63, 58.45, 74.89, 87.12% respectively. Compounds (4c), (4d), (4f) and (4g) has been also shown growth inhibition 49.90, 67.70, 79.45, 91.58%, 31.25, 48.70, 67.93, 82.56%, 33.30, 52.78, 77.80, 88.45% and 29.96, 49.00, 68.12, 83.59% respectively against *Colletotrichum gloeosporioides* fungus at different concentrations 250, 500, 1000 and 2000 μg/mL. Compound (4e) has shown no growth inhibition at all concentrations. Compound (4h) has shown no growth inhibition at lower concentrations. Compound (4h) exhibited 40.50 and 68.54% growth inhibition against *Colletotrichum gloeosporioides* fungus at 1000 μg/mL and 2000 μg/mL concentration respectively. From antifungal activity results, we concluded that compound (4f) was shown to most promising against *R. solani* and compound (4c) was shown to most promising against *C. gloeosporioides*. This result may be due to substitution of hydroxy and methoxy groups on phenyl ring. The box plot and graphical representation of antifungal activity of all compounds against *Rhizoctonia solani* and *Colletotrichum gloeosporioides* were shown in Figs 8–11.

### Antibacterial activity

The optimistic antifungal activity of synthesized compounds (4a-4h) has influenced authors to test further for antibacterial activity. All synthesized compounds (4a-4h) were tested for their *in vitro* antibacterial activity against two bacterial strains *Erwinia cartovora* and *Xanthomonas*

**Table 6. Antifungal activity of substituted dihydropyrimidinones (4a-4h).**

| Compounds | Growth inhibition (%) | | | | | | | |
|---|---|---|---|---|---|---|---|---|
| | Fungi | | | | | | | |
| | *Rhizoctonia solani* (conc.) μg/mL | | | | *Colletotrichum gloeosporioides* (conc.) μg/mL | | | |
| | 250 | 500 | 1000 | 2000 | 250 | 500 | 1000 | 2000 |
| 4a | 61.53 ± 0.90 | 71.15 ± 1.03 | 80.76 ± 2.01 | 86.53 ± 1.08 | 37.89 ± 1.16 | 53.80 ± 0.77 | 78.18 ± 0.16 | 89.78 ± 1.09 |
| 4b | 41.35 ± 0.89 | 62.56 ± 0.72 | 78.13 ± 0.70 | 89.38 ± 1.04 | 35.63 ± 1.19 | 58.45 ± 1.16 | 74.89 ± 1.10 | 87.12 ± 0.96 |
| 4c | 50.87 ± 1.04 | 69.99 ± 0.78 | 80.00 ± 2.67 | 91.13 ± 1.95 | 49.90 ± 1.05 | 67.70 ± 0.50 | 79.45 ± 1.10 | 91.58 ± 1.06 |
| 4d | 40.38 ± 1.53 | 65.38 ± 1.02 | 80.76 ± 1.76 | 90.38 ± 1.42 | 31.25 ± 1.00 | 48.70 ± 1.26 | 67.93 ± 0.98 | 82.56 ± 0.61 |
| 4e | 35.70 ± 1.00 | 54.68 ± 0.37 | 67.30 ± 1.61 | 79.89 ± 2.26 | a | a | a | a |
| 4f | 55.50 ± 1.64 | 70.50 ± 2.45 | 80.98 ± 2.26 | 92.00 ± 1.02 | 33.30 ± 1.59 | 52.78 ± 1.39 | 77.80 ± 0.99 | 88.45 ± 0.61 |
| 4g | 48.27 ± 1.92 | 65.51 ± 1.62 | 75.86 ± 2.41 | 82.75 ± 1.37 | 29.96 ± 1.36 | 49.00 ± 0.71 | 68.12 ± 0.92 | 83.59 ± 0.45 |
| 4h | 60.78 ± 1.84 | 70.58 ± 1.61 | 82.35 ± 1.06 | 90.19 ± 0.64 | a | a | 40.50 ± 1.70 | 68.54 ± 0.91 |

All values are mean ± S.D.

a: No Growth inhibition

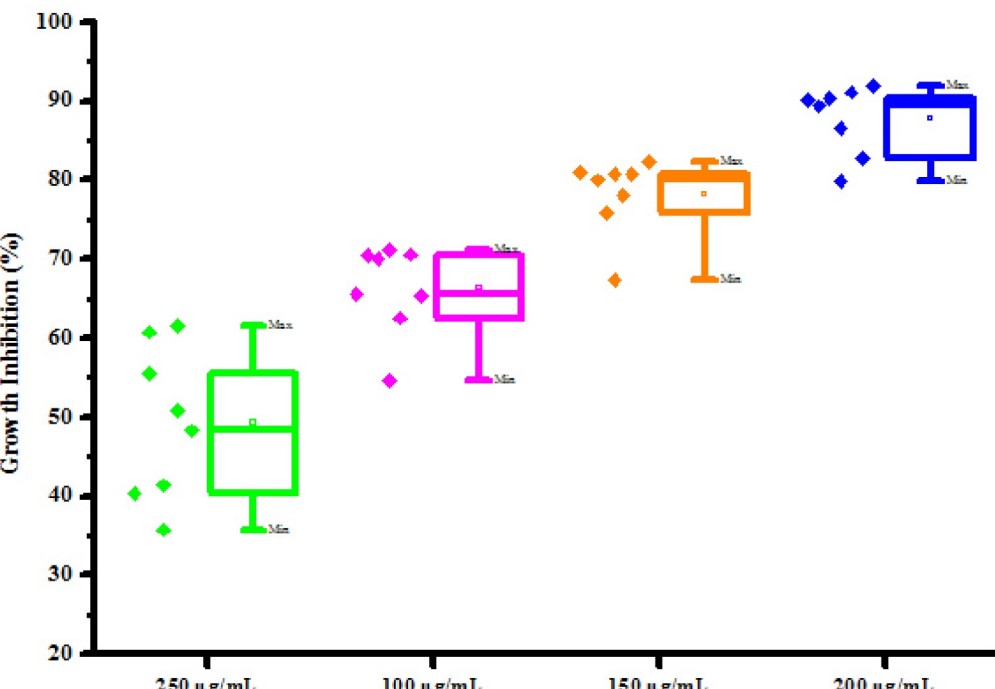

**Fig 8. Box plot of substituted dihyropyrimidinones (4a-4h) against *Rhizoctonia solani*.**

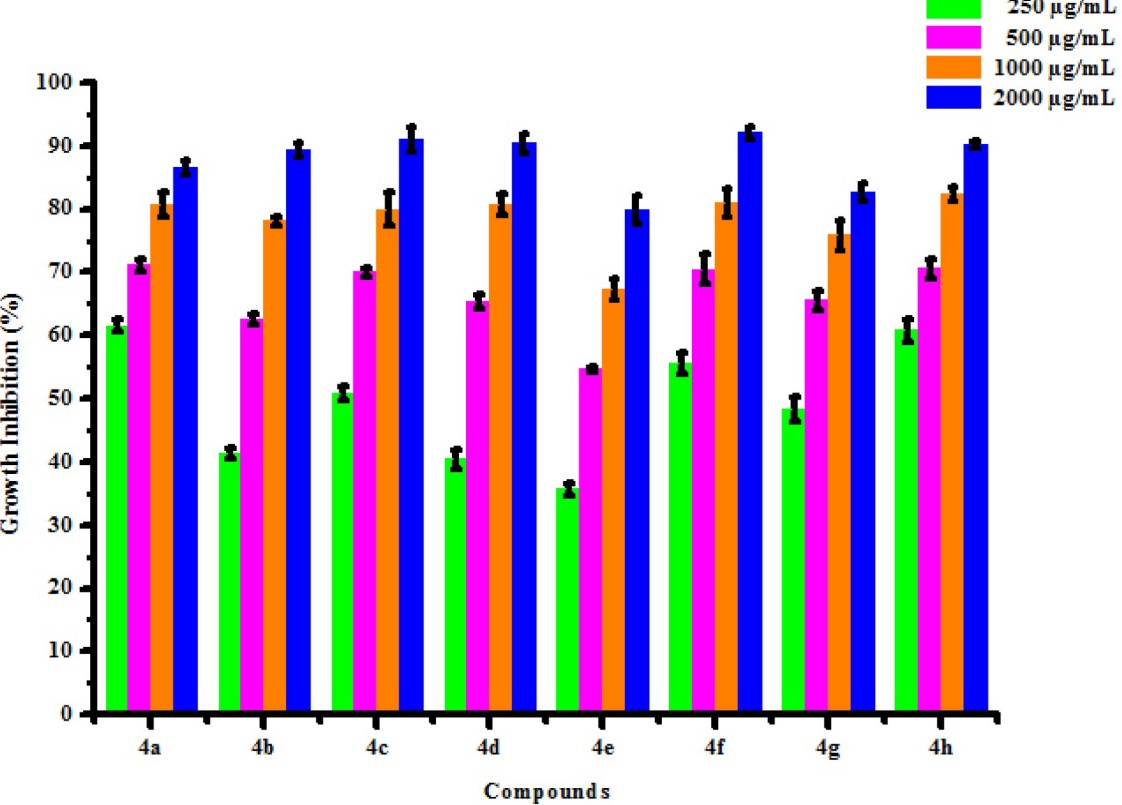

**Fig 9. Antifungal activity of substituted dihyropyrimidinones (4a-4h) against *Rhizoctonia solani*.**

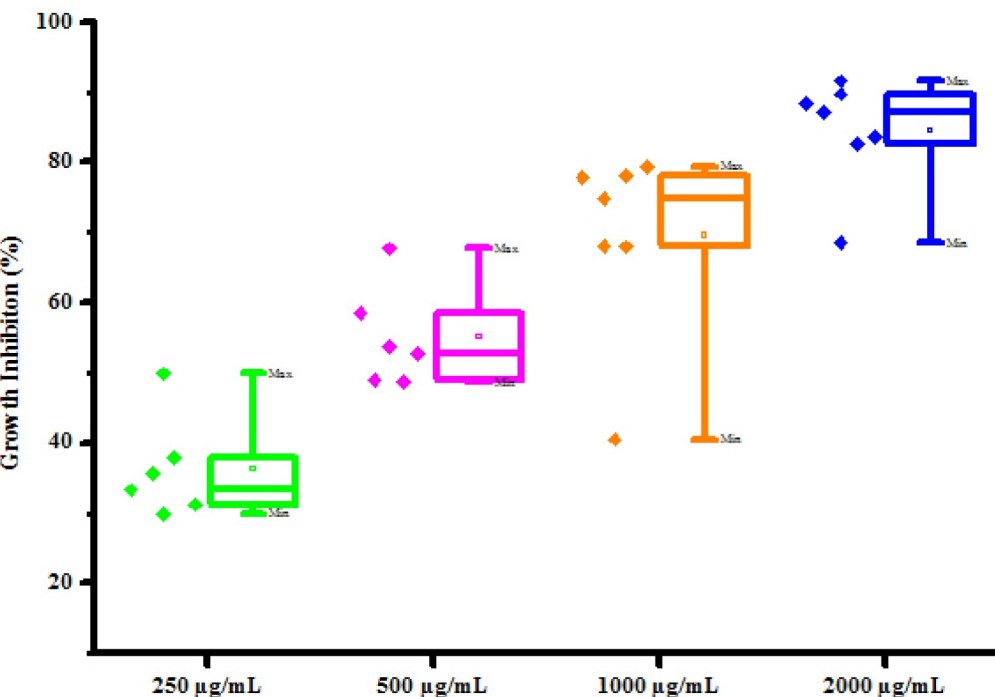

**Fig 10. Box plot of substituted dihyropyrimidinones (4a-4h) against *Colletotrichum gloeosporioides*.**

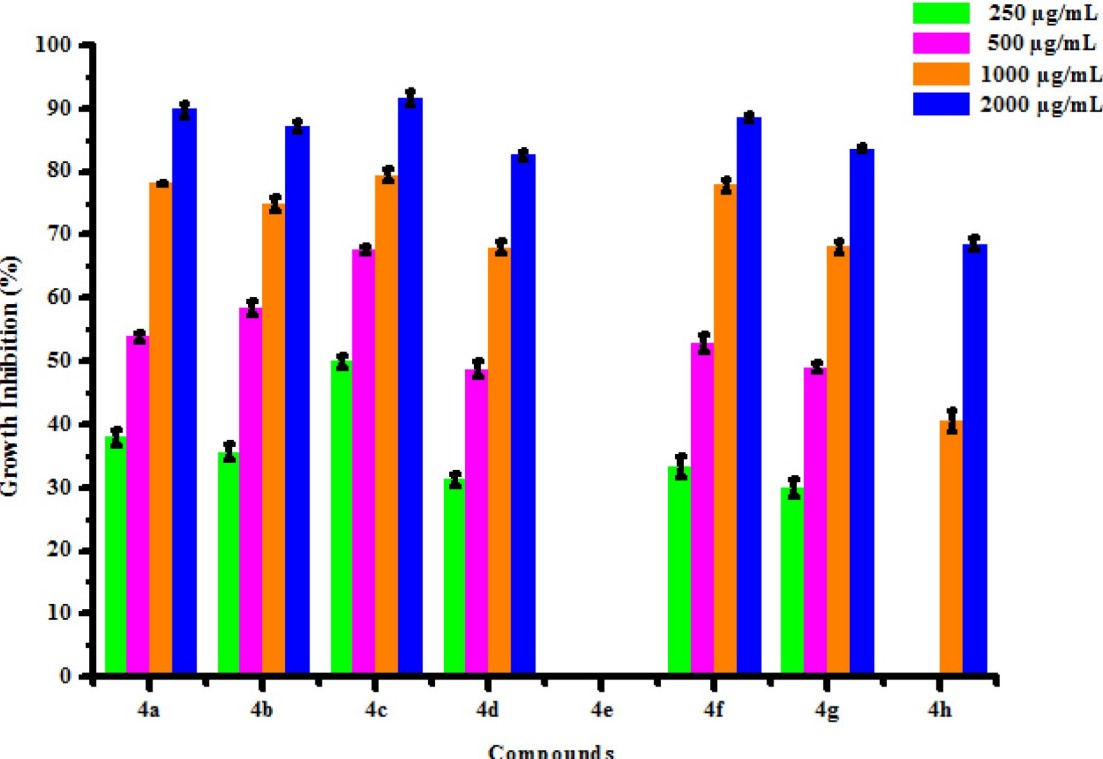

**Fig 11. Antifungal activity of substituted dihyropyrimidinones (4a-4h) against *Colletotrichum gloeosporioides*.**

**Table 7. Antibacterial activity of substituted dihydropyrimidinones (4a-4h).**

| Compounds | Inhibition Zone (mm) | | | | | | | |
|---|---|---|---|---|---|---|---|---|
| | Bacteria | | | | | | | |
| | *Erwinia cartovora* (conc.) µg/mL | | | | *Xanthomonas citri* (conc.) µg/mL | | | |
| | 250 | 500 | 1000 | 2000 | 250 | 500 | 1000 | 2000 |
| 4a | a | a | a | a | a | a | a | a |
| 4b | 0.90 ± 0.07 | 1.50 ± 0.10 | 2.00 ± 0.30 | 3.00 ± 0.40 | 1.40 ± 0.20 | 2.10 ± 0.18 | 3.40 ± 0.20 | 4.10 ± 0.45 |
| 4c | 1.70 ± 0.30 | 2.80 ± 0.35 | 4.00 ± 0.41 | 5.00 ± 0.45 | 0.90 ± 0.07 | 1.50 ± 0.28 | 1.70 ± 0.15 | 2.20 ± 0.47 |
| 4d | a | a | a | a | a | a | a | a |
| 4e | a | a | 0.70 ± 0.07 | 1.10 ± 0.26 | 3.00 ± 0.47 | 5.50 ± 0.30 | 7.60 ± 0.30 | 9.90 ± 0.43 |
| 4f | 1.00 ± 0.18 | 2.00 ± 0.45 | 3.00 ± 0.15 | 5.00 ± 0.55 | 4.00 ± 0.50 | 7.10 ± 0.36 | 9.50 ± 0.26 | 12.0 ± 0.40 |
| 4g | 1.00 ± 0.09 | 2.50 ± 0.35 | 3.00 ± 0.16 | 5.50 ± 0.50 | a | a | 1.10 ± 0.20 | 2.00 ± 0.30 |
| 4h | 1.00 ± 0.12 | 2.00 ± 0.17 | 3.00 ± 0.15 | 4.00 ± 0.45 | 1.10 ± 0.22 | 1.90 ± 0.40 | 3.00 ± 0.26 | 5.00 ± 0.50 |

All values are mean ± S.D.

a: No Inhibition Zone

*citri* by inhibition zone method using DMSO as negative control. The results of antibacterial activity of synthesized compounds were shown in Table 7. No inhibition zone was shown by compounds (4a) and (4d) at all the concentration against *Erwinia cartovora*. Compounds (4b) and (4c) was found active against *Erwinia cartovora* at 250, 500, 1000 and 2000 µg/mL concentrations showing inhibition zone 0.90 mm, 1.50 mm, 2.00 mm, 3.00 mm and 1.70 mm, 2.80 mm, 4.00 mm, 5.00 mm respectively. Compound (4e) has shown no inhibition zone at lower concentrations. Compound (4e) exhibited 0.70 mm and 1.10 mm inhibition zone against *Erwinia cartovora* at 1000 and 2000 µg/mL concentration respectively. Compounds (4f), (4g) and (4h) was found active against *Erwinia cartovora* at 250, 500, 1000 and 2000 µg/mL concentrations showing inhibition zone 1.00 mm, 2.00 mm, 3.00 mm, 5.00 mm, 1.00 mm, 2.50 mm, 3.00 mm, 5.50 mm and 1.00 mm, 2.00 mm, 3.00 mm, 4.00 mm respectively. Compounds (4a) and (4d) have shown no inhibition zone at all the concentrations against *Xanthomonas citri*. Compounds (4b) and (4c) was found active against *Xanthomonas citri* at 250, 500, 1000 and 2000 µg/mL concentrations showing inhibition zone 1.40 mm, 2.10 mm, 3.40 mm, 4.10 mm and 0.90 mm, 1.50 mm, 1.70 mm, 2.20 mm respectively. Compounds (4e) and (4f) was found active against *Xanthomonas citri* at 250, 500, 1000 and 2000 µg/mL concentrations showing inhibition zone 3.00 mm, 5.50 mm, 7.60 mm, 9.90 mm and 4.00 mm, 7.10 mm, 9.50 mm, 12.00 mm respectively. Compound (4g) has shown no inhibition zone at lower concentrations. Compound (4g) exhibited 1.10 mm and 2.00 mm inhibition zone against *Xanthomonas citri* at 1000 and 2000 µg/mL concentration respectively. Compounds (4h) was found active against *Xanthomonas citri* at 250, 500, 1000 and 2000 µg/mL concentrations showing inhibition zone 1.10 mm, 1.90 mm, 3.00 mm, 5.00 mm respectively. Maximum *Erwinia cartovora* growth was inhibited by compounds (4g) showing inhibition zone 1.00–5.50 mm. Maximum *Xanthomonas citri* growth was inhibited by compounds (4f) showing inhibition zone 4.00–12.00 mm. This inhibition may be due to presence of chloro and hydroxy substitution on phenyl groups. The box plot and graphical representation of antibacterial activity of all compounds against *Erwinia cartovora* and *Xanthomonas citri* were shown in Figs 12–15.

## Characterization data of selected compounds

**Methyl 4-(2-hydroxyphenyl)-6-methyl-2-oxo-1,2,3,4-tetrahydropyrimidine-5-carboxylate (4a):** [1]H NMR (400 MHz, DMSO-$d_6$): $\delta$ 1.75 (s, 3H, **CH₃**); 3.71 (s, 3H, CO**OCH₃**); 4.51 (s, 1H, **OH**); 6.73–7.19 (m, *J* = 8 Hz, 4H, **Ar-H**); 7.46 (s, 1H, **NH**); 7.69 (s, 1H, **NH**)

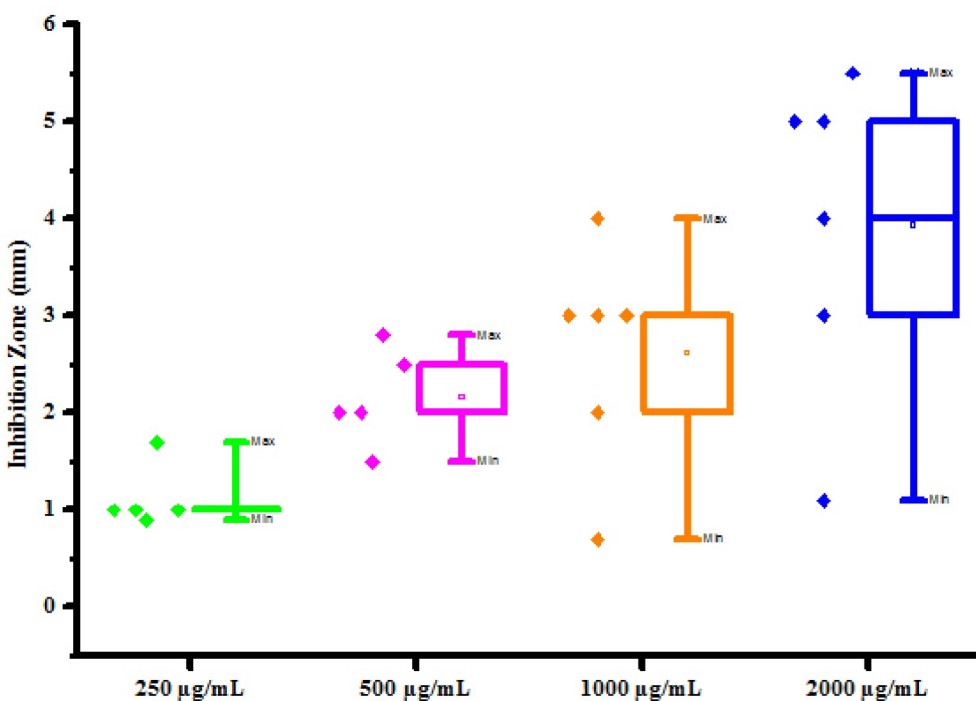

**Fig 12. Box plot of substituted dihyropyrimidinones (4a-4h) against *Erwina cartovora*.**

**Methyl 6-(4-methoxyphenyl)-4-methyl-2-oxo-1, 2-dihydropyrimidine-5-carboxylate (4b):** [1]H NMR (400 MHz, DMSO-$d_6$): $\delta$ 2.25 (s, 3H, **CH$_3$**); 3.72 (s, 3H, Ar-**OCH$_3$**); 3.51 (s, 3H, CO**OCH$_3$**); 6.57–7.66 (m, $J$ = 8 Hz, 4H, **Ar-H**); 5.11 (s, 1H, **NH**); 9.16 (s, 1H, **NH**)

**Methyl 4-(3,4-dimethoxyphenyl)-6-methyl-2-oxo-1,2,3,4-tetrahydropyrimidine-5-carboxylate (4c):** [1]H NMR (400 MHz, DMSO-$d_6$): $\delta$ 2.51 (s, 3H, **CH$_3$**); 3.82 (s, 3H, Ar-**OCH$_3$**); 3.86 (s, 3H, Ar-**OCH$_3$**); 7.13–7.16 (d, $J$ = 12 Hz, 1H, **Ar-H**); 7.37–7.38 (d, $J$ = 4 Hz, 1H, **Ar-H**); 7.53–7.55 (m, $J$ = 8 Hz, 1H, **Ar-H**); 5.48 (s, 1H, **NH**); 9.83 (s, 1H, **NH**)

**Methyl 6-(4-chlorophenyl)-4-methyl-2-oxo-1,2-dihydropyrimidine-5-carboxylate (4d):** IR ($\nu_{max}$ cm$^{-1}$) (neat): 3415.3 (**NH**), 3325.7 (**NH**), 1680.5 (**C = O**), 1591.5 (**C = C**, aromatic), 761.0 (**C-Cl**)

**Methyl 6-(4-bromophenyl)-4-methyl-2-oxo-1,2-dihydropyrimidine-5-carboxylate (4e):** IR ($\nu_{max}$ cm$^{-1}$) (neat): 3436.6 (**NH**), 3307.5 (**NH**), 1696.2 (**C = O**), 1587.8 (**C = C**, aromatic), 811.7 (**C-Br**)

**Methyl 6-(3-hydroxyphenyl)-4-methyl-2-oxo-1,2-dihydropyrimidine-5-carboxylate (4f):** IR ($\nu_{max}$ cm$^{-1}$) (neat): 3227.7 (**NH**), 3106.4 (**NH**), 3372.4 (**OH**), 1714.0 (**C = O**), 1487.9 (**C = C**, aromatic)

**Methyl 6-(2-chlorophenyl)-4-methyl-2-oxo-1,2-dihydropyrimidine-5-carboxylate (4g):** [1]H NMR (400 MHz, DMSO-$d_6$): $\delta$ 2.31 (s, 3H, **CH$_3$**); 3.46 (s, 3H, CO**OCH$_3$**); 7.20–7.36 (m, $J$ = 8 Hz, 4H, **Ar-H**); 5.65 (s, 1H, **NH**); 9.27 (s, 1H, **NH**)

**Methyl 4-methyl-2-oxo-6-(p-tolyl)-1,2-dihydropyrimidine-5-carboxylate (4h):** IR ($\nu_{max}$ cm$^{-1}$) (neat): 3484.4 (NH), 3335.7 (NH), 1740.5 (C = O), 1651.7 (C = C, aromatic)

## Conclusions

We have reported a facile one-pot three component synthesis of substituted dihydropyrimidinones derivatives (4a-4h) by condensation of substituted aldehyde, methyl acetoacetate and

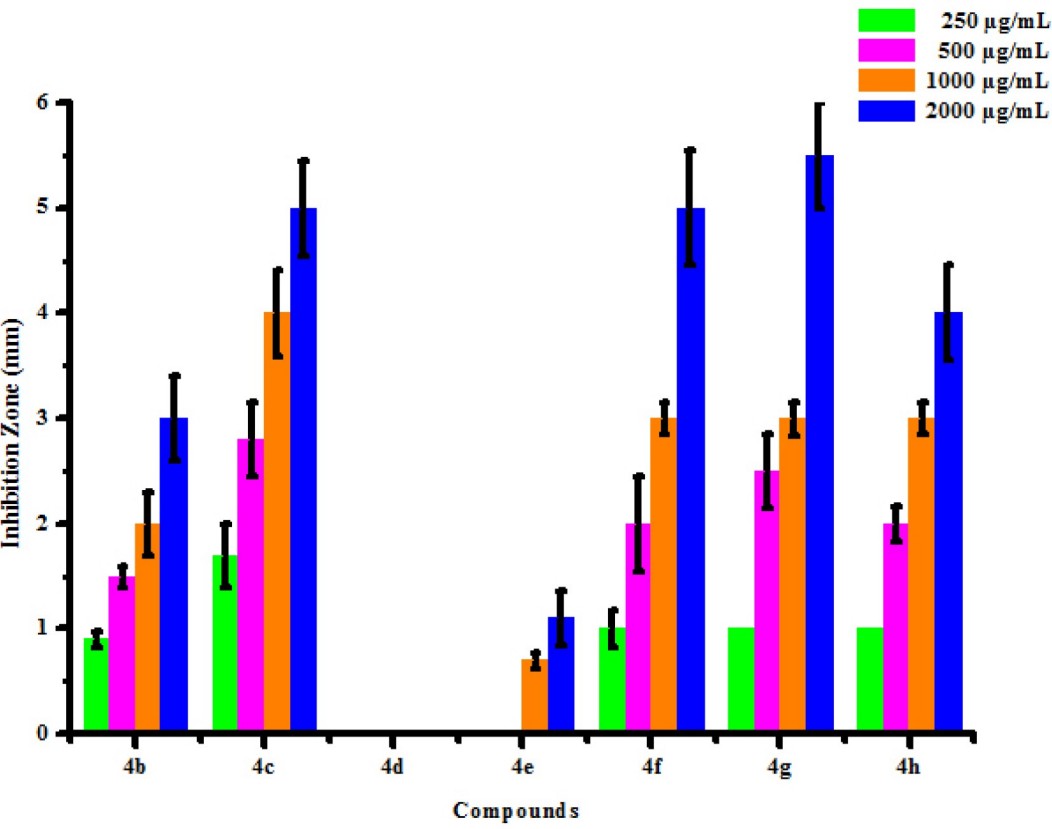

**Fig 13. Antibacterial activity of substituted dihyropyrimidinones (4a-4h) against *Erwina cartovora*.**

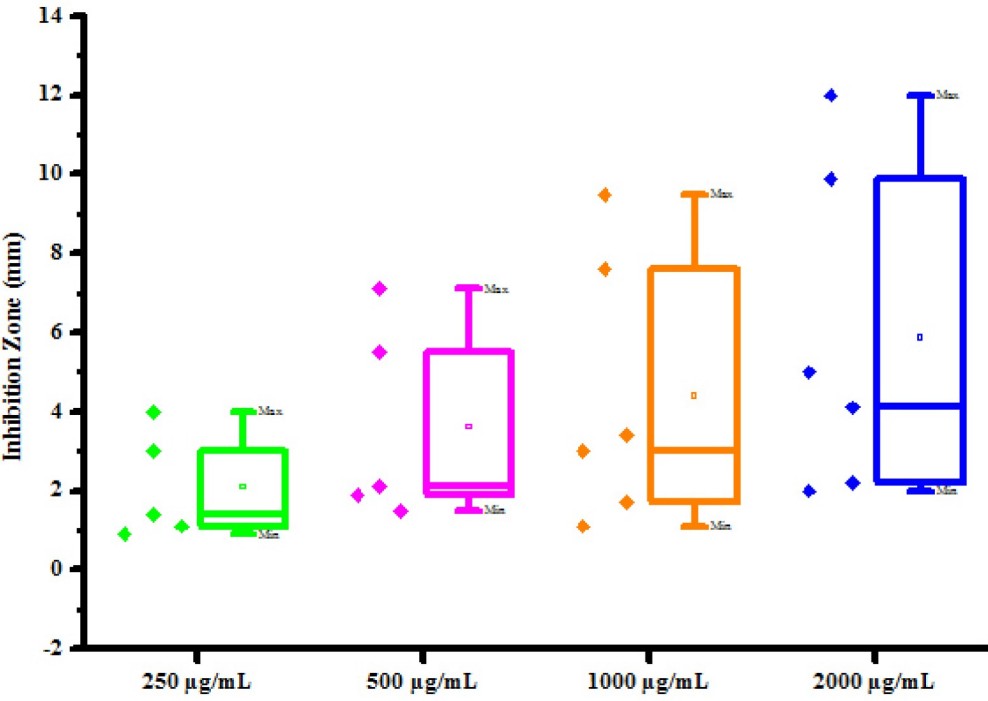

**Fig 14. Box plot of substituted dihyropyrimidinones (4a-4h) against *Xanthomonas citri*.**

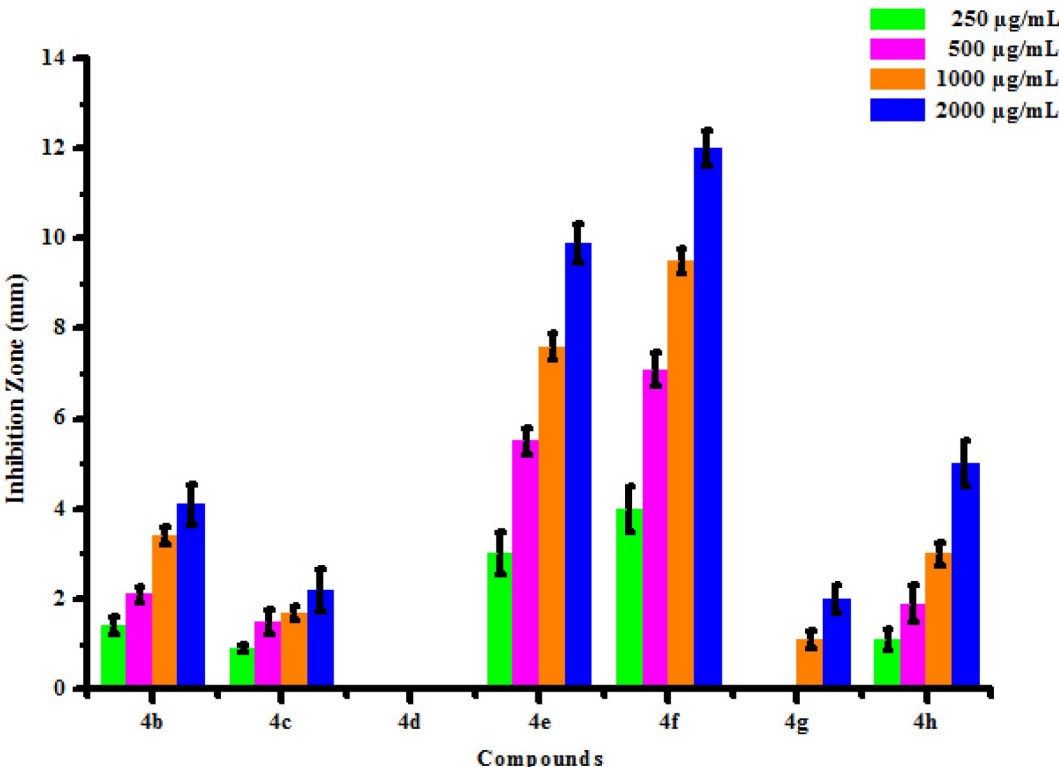

**Fig 15. Antibacterial activity of substituted dihydropyrimidinones (4a-4h) against *Xanthomonas citri*.**

urea at room temperature in presence of *Cocos nucifera* L. juice, *Solanum lycopersicum* L. juice and *Citrus limetta* juice in excellent yields. The current procedure offers many advantages such as simple and efficient catalytic system, simple work-up, no use of hazardous solvents, cheap and products were obtained in good to excellent yields. Moreover, all products were obtained through simple filtration with no need for column chromatography, which reduces the waste as well as environmental pollution. All synthesized compounds (4a-4h) were also evaluated for their bioevaluation in terms of herbicidal activity against *Raphanus sativus* L. (Radish) seeds, antifungal activity against *Rhizoctonia solani &Colletotrichum gloeosporioides* and antibacterial activity against *Erwinia cartovora* and *Xanthomonas citri*. Based on activity data, it can be concluded that some of synthesized compounds exhibited good activity due to substitution of chloro, hydroxy and methoxy substitution on phenyl ring. We also conclude that current protocol will provide great utility in the synthesis of other heterocyclic compounds in the near future.

## Supporting information

**S1 Fig.**
(DOCX)

**S2 Fig.**
(DOCX)

**S3 Fig.**
(DOCX)

**S4 Fig.**
(DOCX)

**S5 Fig.**
(DOCX)

**S6 Fig.**
(DOCX)

**S7 Fig.**
(DOCX)

**S8 Fig.**
(DOCX)

**S9 Fig.**
(DOCX)

**S10 Fig.**
(DOCX)

**S11 Fig.**
(DOCX)

**S12 Fig.**
(DOCX)

**S13 Fig.**
(DOCX)

**S14 Fig.**
(DOCX)

**S15 Fig.**
(DOCX)

**S16 Fig.**
(DOCX)

**S17 Fig.**
(DOCX)

**S18 Fig.**
(DOCX)

**S19 Fig.**
(DOCX)

**S20 Fig.**
(DOCX)

**S21 Fig.**
(DOCX)

**S22 Fig.**
(DOCX)

**S23 Fig.**
(DOCX)

**S1 Table.**
(DOCX)

**S2 Table.**
(DOCX)

**S3 Table.**
(DOCX)

**Graphical abstract.**
(TIF)

## Acknowledgments

The authors are thankful to the Department of Chemistry, Chaudhary Charan Singh Haryana Agricultural University, Hisar for providing the necessary facilities. Authors are also thankful to SAIF, Punjab University Chandigarh, for providing analytical facilities for characterization of compounds.

## Author Contributions

**Supervision:** Rajvir Singh, Ram Prakash.

**Visualization:** Suman Sangwan.

**Writing – original draft:** Susheel Gulati.

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
