## [Decision Letter · Decision Letter 0]

7 Jul 2020

PONE-D-20-15280

One-Pot Three Component Synthesis of Substituted Dihydropyrimidinones Using Fruit Juices as Biocatalyst and their Biological Studies

PLOS ONE

Dear Dr. Gulati,

Thank you for submitting your manuscript to PLOS ONE. After careful consideration, we feel that it has merit but does not fully meet PLOS ONE’s publication criteria as it currently stands. Therefore, we invite you to submit a revised version of the manuscript that addresses the points raised during the review process.

The manuscript was reviewed by three experts who have differing opinions, including one who recommended rejection. The revised manuscript should address all the comments and feedback provided by the reviewers. To summarise, here are the key points:

Simplify the abstractCite appropriate scientific literature including the examples recommended by the reviewersCorrect solvent free to ‘under aqueous conditions’Carry out quantitative measurement of natural acids and proteins as requested by reviewer 1Discuss the nature of the catalyst- ascorbic acid, other acids, metal ions etc.Follow the literature precedents for discussion on the mechanismWrite clear experimental procedures to enable reproducibility by othersProvide detailed characterization of compounds including copies of NMR spectra to confirm compound purity

We look forward to receiving your revised manuscript.

Kind regards,

A Ganesan

Academic Editor

PLOS ONE

Journal Requirements:

3. We noticed you have some minor occurrence of overlapping text with previous publications, which needs to be addressed. In your revision ensure you cite all your sources (including your own works), and quote or rephrase any duplicated text outside the methods section. Further consideration is dependent on these concerns being addressed.

4. Please amend your Financial disclosure statement to declare sources of funding, or state that the authors received no specific funding.

Thank you for stating the following in the Acknowledgments Section of your manuscript:

"Financial assistance from Department of Science and Technology (DST), New Delhi, India is gratefully acknowledged."

"NO"

5. We note that this submission includes NMR spectroscopy data. We would recommend that you include the following information in your methods section or as Supporting Information files:

1) The make/source of the NMR instrument used in your study, as well as the magnetic field strength. For each individual experiment, please also list: the nucleus being measured; the sample concentration; the solvent in which the sample is dissolved and if solvent signal suppression was used; the reference standard and the temperature.

2) A list of the chemical shifts for all compounds characterised by NMR spectroscopy, specifying, where relevant: the chemical shift (δ), the multiplicity and the coupling constants (in Hz), for the appropriate nuclei used for assignment.

3)The full integrated NMR spectrum, clearly labelled with the compound name and chemical structure.

We also strongly encourage authors to provide primary NMR data files, in particular for new compounds which have not been characterised in the existing literature. Authors should provide the acquisition data, FID files and processing parameters for each experiment, clearly labelled with the compound name and identifier, as well as a structure file for each provided dataset. See our list of recommended repositories here: https://journals.plos.org/plosone/s/recommended-repositories

6. We note that Graphical Abstract in your submission contain copyrighted images. All PLOS content is published under the Creative Commons Attribution License (CC BY 4.0), which means that the manuscript, images, and Supporting Information files will be freely available online, and any third party is permitted to access, download, copy, distribute, and use these materials in any way, even commercially, with proper attribution. For more information, see our copyright guidelines: http://journals.plos.org/plosone/s/licenses-and-copyright.

6. 1.         You may seek permission from the original copyright holder of Graphical Abstract to publish the content specifically under the CC BY 4.0 license.

6.2.    If you are unable to obtain permission from the original copyright holder to publish these figures under the CC BY 4.0 license or if the copyright holder’s requirements are incompatible with the CC BY 4.0 license, please either i) remove the figure or ii) supply a replacement figure that complies with the CC BY 4.0 license. Please check copyright information on all replacement figures and update the figure caption with source information. If applicable, please specify in the figure caption text when a figure is similar but not identical to the original image and is therefore for illustrative purposes only.

7. In your Data Availability statement, you have not specified where the minimal data set underlying the results described in your manuscript can be found. PLOS defines a study's minimal data set as the underlying data used to reach the conclusions drawn in the manuscript and any additional data required to replicate the reported study findings in their entirety. All PLOS journals require that the minimal data set be made fully available. For more information about our data policy, please see http://journals.plos.org/plosone/s/data-availability.

Reviewers' comments:

Reviewer's Responses to Questions

**Comments to the Author**

1. Is the manuscript technically sound, and do the data support the conclusions?

Reviewer #1: No

Reviewer #2: Yes

Reviewer #3: Yes

2. Has the statistical analysis been performed appropriately and rigorously? 

Reviewer #1: No

Reviewer #2: Yes

Reviewer #3: N/A

3. Have the authors made all data underlying the findings in their manuscript fully available?

Reviewer #1: No

Reviewer #2: Yes

Reviewer #3: Yes

4. Is the manuscript presented in an intelligible fashion and written in standard English?

Reviewer #1: No

Reviewer #2: Yes

Reviewer #3: Yes

5. Review Comments to the Author

Reviewer #1: Substituted Dihydropyrimidinones are very well known from high biological activity. Common synthetic method leading to target product is based on one-pot three - components reaction. Several aspects of this reaction were already discussed within literature. The authors used as a solvent for reaction fruit juices viz. Cocos nucifera L., Solanum lycopersicum L., Citrus limetta. Model experiments on 4-hydroxy-3-methoxybenzaldehyde, methyl acetoacetate and urea were performed in Cocos nucifera L. juice. The juices were used as prepared and not characterized. The protein contents, the amounts of natural acids was not determined what makes this procedure completely unrepeatable. Although the authors claims in graphical abstract that natural acids participate in reaction that was not validated.

The results of model experiments showed in Table 1 and 2 are confusing to readers since respective experimental procedures are entirely different. It seems to reader that the amount of juice was changed from 1 to 2.5 mL (Table 1) and rom 4 to 10 ml (Table 2). This is just the dilution effect on the reaction course not catalytic effect. The product which should be obtained in reaction is missed in experimental part. The synthetic procedures are full of mistakes and misunderstandings. In method B on page 33 reaction mixture was stirred at room temperature and then cooled to room temperature. Compounds provided in Table 3 were characterized by melting points and 1H NMR what is far away from Journal requirements. The purity of all substances was not determined what do not fulfill requirements of the Journal. The catalytic effect of juices was only postulated and was not verified. The contents of Table 4 is difficult to understand since the experimental procedure is not included in manuscript body. The possible mechanism of reaction discussed in Scheme 2 should be based on sole literature data not speculations. Biological dates were collected for compounds of undefined purity.

Reviewer #2: Authors have described a new and facile one-pot three component synthetic protocol for substituted dihydropyrimidinones derivatives via reaction of equimolar substituted aldehydes, methyl acetoacetate and urea in presence of nature derived catalyst viz. Cocos nucifera L. juice, Solanum lycopersicum L. juice and Citrus limetta juice, commonly known as coconut juice, tomato juice and musambi juice, respectively, at room temperature.

This reviewer doesn’t have much in terms of criticism for the described work which has been very well explored in the literature before; one of the most explored reaction under dozens of reaction conditions.

The major comment at the outset is to decrease the verbose descriptions in the abstract; it reads more like an introduction and experimental presently; no need to mention TLC, or characterization details, mmol scale and in fact with proper referencing, they can avoid lots of duplication of data on known compounds. Secondly, authors have described properly in conclusion but to say that it is a solvent-free reaction is not correct as juices are mostly water. So, aqueous media is the correct description.

I that context, I think it would be prudent to have some quality references while describing the reaction as most of the citations are poor quality and not from original works. As an example, this reviewer could easily find 2 examples for solvent-free and aqueous medium reaction, respectively on the title compounds, as shown below:

Microwave-Assisted High-Speed Parallel Synthesis of 4-Aryl-Dihydro-pyrimidin-2(1H)-ones using a Solventless Biginelli Condensation Protocol. Synthesis, 1799-1803 (1999)

and

Biginelli Reaction in Aqueous Medium: A Greener and Sustainable Approach to Substituted 3,4-Dihydropyrimidin-2(1H)-ones. Tetrahedron Lett., 48, 7343 (2007). Similar types of citations should be offered in the introduction section and like most of the paper, description need to be curtailed enormously. Authors have not cited a relevant paper that describes the utility of juices, “Exploring the utility of fruit juices as green medium for Biginelli reaction”, December 2013, Research Journal of Pharmaceutical, Biological and Chemical Sciences 5(5) :444-449 and related papers by TANAY PRAMANIK, SIMARJIT KAUR PADAN on this topic.

Reviewer #3: The authors here display an important study that used some natural juices as catalysts to attain Bignilli Reaction

and study the Herbicidal activity and antibacterial activity of the obtained compounds

The results show that Cocos nucifera L. juice is the best catalyst for the used time 10 min.

But I am surprising that author put in graphical abstract some acid that may be mean the component of juices

this is not logic if the catalytic activity of Cocos nucifera L. juice due to Ascorbic acid, therefor the results of ascorbic acid should be put in table 2 which present in the following ref.

Ascorbic acid-Catalyzed One-Pot Three-Component Biginelli Reaction: A Practical and Green Approach towards Synthesis of 3,4-dihydropyrimidin- 2(1H)-ones/thiones

Journal Name: Letters in Organic Chemistry

Volume 10 , Issue 7 , 2013

But Cocos nucifera L. juice contain many sugers, many vitamins includs Vit. C (Ascorbic acid) and many metal ion that may be make a synergistic action that give this catalytic activity.

Therefore authors should remove the acids that present in Graphical Abstract.

also, authors should submit NMR Charts for the synthesized compounds to ensure the purity.

6. PLOS authors have the option to publish the peer review history of their article (what does this mean?). If published, this will include your full peer review and any attached files.

Reviewer #1: No

Reviewer #2: No

Reviewer #3: **Yes: **Tamer S. Saleh

---

## [Author Response · Author response to Decision Letter 0]

10 Jul 2020

R/Sir all the suggestions according to reviewer comments incorporated successfully in revised manuscript. Thanks to editorial staff and reviewer their timely support and necessary suggestions for making this manuscript fruitful. All suggestions were shown in manuscript by track change mode.

Comments from the Editors and Reviewers:

Comments to the Author

Simplify the abstract

Answer to Reviewer:

Authors simplify the abstract according to reviewer suggestions in revised manuscript.

Comments to the Author

Cite appropriate scientific literature including the examples recommended by the reviewers

Answer to Reviewer:

Authors cited the appropriate scientific literature in the manuscript. For e.g.

1. Patil S, Jadhav SD, Deshmukh MB. Natural acid catalyzed multi-component reactions as a green approach. Arch. Apll. Sci. Res., 2011 3(1):203-208.

2. Tamuli KJ, Dutta D, Nath S, Bordoloi M. A Greener and Facile Synthesis of Imidazole and Dihydropyrimidine Derivatives under Solvent‐Free Condition Using Nature‐Derived Catalyst. Chemistry Select. 2017 2(26):7787-7791.

3. Fonseca AM, Monte FJ, Maria da Conceição F, de Mattos MC, Cordell GA, Braz-Filho R, Lemos TL. Coconut water (Cocos nucifera L.)—A new biocatalyst system for organic synthesis. J. Mol. Catal. B Enzym. 2009 57(1-4):78-82.

4. Nazeruddin GM, Shaikh YI. Tamarind juice catalyzed one pot synthesis of dihydropyrimidinone and thione under ultrasound irradiation at ambient conditions: A green approach. Der Pharmacia Sinica., 2014 5(6):64-68.

5. Ramu E, Kotra V, Bansal N, Varala R, Adapa SR. Green approach for the efficient synthesis of Biginelli compounds promoted by citric acid under solvent-free conditions. Rasayan Journal of Chem., 2008 1(1):188-194.

Comments to the Author

Correct solvent free to ‘under aqueous conditions’

Answer to Reviewer:

Authors replaced solvent-free to ‘under aqueous conditions’ in revised manuscript.

Comments to the Author

Carry out quantitative measurement of natural acids and proteins as requested by reviewer 1

Answer to Reviewer:

Authors in their experiment not quantify natural acids and proteins.

Comments to the Author

Discuss the nature of the catalyst- ascorbic acid, other acids, metal ions etc.

Answer to Reviewer:

Authors used nature derived catalyst viz. Cocos nucifera L. juice, Solanum lycopersicum L. juice and Citrus limetta juice in their study. All these catalyst contain various organic acids mainly ascorbic acid, citric acid and malic acid etc. All these acids are weak organic acid. Ascorbic acid is an organic compounds called hexuronic acid and it dissolves well in water to give mildly acidic solutions. It is mild reducing agent.

Comments to the Author

Follow the literature precedents for discussion on the mechanism

Answer to Reviewer:

Authors cited the literature for discussion on the mechanism in revised manuscript.

Comments to the Author

Write clear experimental procedures to enable reproducibility by others

Answer to Reviewer:

Authors improved the experimental procedure in revised manuscript. Now the experimental procedure becomes clearer.

Comments to the Author

Provide detailed characterization of compounds including copies of NMR spectra to confirm compound purity

Answer to Reviewer:

Authors provided detailed characterization (IHNMR and FTIR) of some selected compounds to confirm compound purity.

Compounds name and structure also provided in NMR and FTIR chart file.

Authors revised the manuscript according to PLOS ONE style.

Authors deleted the funding information in the Acknowledgments section in revised manuscript.

Authors revised the graphical abstract according to reviewer suggestions.

Authors successfully added the details of colleague who edited the manuscript in revised manuscript.

Authors successfully added the financial disclosure statement “Authors receive no specific funding” in revised manuscript.

Authors successfully deleted the copy right images from the graphical abstract and revised the graphical abstract according to reviewer suggestions in revised manuscript.

Authors successfully added Data availability statement in revised manuscript.

Authors successfully uploaded minimal underlying data set as either Supporting Information files in revised manuscript.

Authors thoroughly copyedit the manuscript for language usage, spelling, and grammar.

In manuscript all the graph made by OriginPro 8 software and standard deviation was also included in graphical representation to make manuscript more scientifically.

All the structure in manuscript made by using ChemDraw Ultra 12.0 software.

Manuscript is now accordance to PLOS ONE style.

Thanks to the editorial board and reviewers for necessary suggestions regarding the manuscript.

---

## [Decision Letter · Decision Letter 1]

6 Aug 2020

PONE-D-20-15280R1

One-Pot Three Component Synthesis of Substituted Dihydropyrimidinones Using Fruit Juices as Biocatalyst and their Biological Studies

PLOS ONE

Dear Dr. Gulati

Thank you for submitting your manuscript to PLOS ONE. After careful consideration, we feel that it has merit but does not fully meet PLOS ONE’s publication criteria as it currently stands. Therefore, we invite you to submit a revised version of the manuscript that addresses the points raised during the review process.

One reviewer has commented on the revised manuscript, and has suggested some additional changes with regards to the references. These recommendations should be followed.

We look forward to receiving your revised manuscript.

Kind regards,

A Ganesan

Academic Editor

PLOS ONE

Reviewers' comments:

Reviewer's Responses to Questions

**Comments to the Author**

1. If the authors have adequately addressed your comments raised in a previous round of review and you feel that this manuscript is now acceptable for publication, you may indicate that here to bypass the “Comments to the Author” section, enter your conflict of interest statement in the “Confidential to Editor” section, and submit your "Accept" recommendation.

Reviewer #2: (No Response)

2. Is the manuscript technically sound, and do the data support the conclusions?

Reviewer #2: Partly

3. Has the statistical analysis been performed appropriately and rigorously? 

Reviewer #2: N/A

4. Have the authors made all data underlying the findings in their manuscript fully available?

Reviewer #2: Yes

5. Is the manuscript presented in an intelligible fashion and written in standard English?

Reviewer #2: (No Response)

6. Review Comments to the Author

Reviewer #2: Although it has improved fair bit but the suggested quality and original references in the introduction section have not been incorporated and the last three references now added are of poor quality. I reproduce what I suggested originally, again:

As an example, this reviewer could easily find 2 examples for solvent-free and aqueous medium reaction, respectively on the title compounds, as shown below:

Microwave-Assisted High-Speed Parallel Synthesis of 4-Aryl-Dihydro-pyrimidin-2(1H)-ones using a Solventless Biginelli Condensation Protocol. Synthesis, 1799-1803 (1999)

and

Biginelli Reaction in Aqueous Medium: A Greener and Sustainable Approach to Substituted 3,4-Dihydropyrimidin-2(1H)-ones. Tetrahedron Lett., 48, 7343 (2007). Similar types of citations should be offered in the introduction section and like most of the paper, description need to be curtailed enormously. Authors have not cited a relevant paper either that describes the utility of juices, “Exploring the utility of fruit juices as green medium for Biginelli reaction”, December 2013, Research Journal of Pharmaceutical, Biological and Chemical Sciences 5(5) :444-449 and related papers by TANAY PRAMANIK, SIMARJIT KAUR PADAN on this topic.

7. PLOS authors have the option to publish the peer review history of their article (what does this mean?). If published, this will include your full peer review and any attached files.

Reviewer #2: No

---

## [Author Response · Author response to Decision Letter 1]

6 Aug 2020

R/Sir all the suggestions according to reviewer comments incorporated successfully in revised (R2 [PONE-D-20-15280R1]) manuscript. Thanks to editorial staff and reviewer for their timely support and necessary suggestions for making revised manuscript more fruitful. All suggestions were shown in manuscript by track change mode. Respond to reviewer were shown in Blue colour.

Comments from the Editors and Reviewers:

Comments to the Author

Simplify the abstract

Answer to Reviewer:

Authors simplify the abstract according to reviewer suggestions in revised manuscript.

Comments to the Author

Cite appropriate scientific literature including the examples recommended by the reviewers

Answer to Reviewer:

Authors cited the appropriate scientific literature in the manuscript. For e.g.

1. Patil S, Jadhav SD, Deshmukh MB. Natural acid catalyzed multi-component reactions as a green approach. Arch. Apll. Sci. Res., 2011 3(1):203-208.

2. Tamuli KJ, Dutta D, Nath S, Bordoloi M. A Greener and Facile Synthesis of Imidazole and Dihydropyrimidine Derivatives under Solvent‐Free Condition Using Nature‐Derived Catalyst. Chemistry Select. 2017 2(26):7787-7791.

3. Fonseca AM, Monte FJ, Maria da Conceição F, de Mattos MC, Cordell GA, Braz-Filho R, Lemos TL. Coconut water (Cocos nucifera L.)—A new biocatalyst system for organic synthesis. J. Mol. Catal. B Enzym. 2009 57(1-4):78-82.

4. Nazeruddin GM, Shaikh YI. Tamarind juice catalyzed one pot synthesis of dihydropyrimidinone and thione under ultrasound irradiation at ambient conditions: A green approach. Der Pharmacia Sinica., 2014 5(6):64-68.

5. Ramu E, Kotra V, Bansal N, Varala R, Adapa SR. Green approach for the efficient synthesis of Biginelli compounds promoted by citric acid under solvent-free conditions. Rasayan Journal of Chem., 2008 1(1):188-194.

Comments to the Author

Correct solvent free to ‘under aqueous conditions’

Answer to Reviewer:

Authors replaced solvent-free to ‘under aqueous conditions’ in revised manuscript.

Comments to the Author

Carry out quantitative measurement of natural acids and proteins as requested by reviewer 1

Answer to Reviewer:

Authors in their experiment not quantify natural acids and proteins.

Comments to the Author

Discuss the nature of the catalyst- ascorbic acid, other acids, metal ions etc.

Answer to Reviewer:

Authors used nature derived catalyst viz. Cocos nucifera L. juice, Solanum lycopersicum L. juice and Citrus limetta juice in their study. All these catalyst contain various organic acids mainly ascorbic acid, citric acid and malic acid etc. All these acids are weak organic acid. Ascorbic acid is an organic compounds called hexuronic acid and it dissolves well in water to give mildly acidic solutions. It is mild reducing agent.

Comments to the Author

Follow the literature precedents for discussion on the mechanism

Answer to Reviewer:

Authors cited the literature for discussion on the mechanism in revised manuscript.

Comments to the Author

Write clear experimental procedures to enable reproducibility by others

Answer to Reviewer:

Authors improved the experimental procedure in revised manuscript. Now the experimental procedure becomes clearer.

Comments to the Author

Provide detailed characterization of compounds including copies of NMR spectra to confirm compound purity

Answer to Reviewer:

Authors provided detailed characterization (IHNMR and FTIR) of some selected compounds to confirm compound purity.

Compounds name and structure also provided in NMR and FTIR chart file.

Authors revised the manuscript according to PLOS ONE style.

Authors deleted the funding information in the Acknowledgments section in revised manuscript.

Authors revised the graphical abstract according to reviewer suggestions.

Authors successfully added the details of colleague who edited the manuscript in revised manuscript.

Authors successfully added the financial disclosure statement “Authors receive no specific funding” in revised manuscript.

Authors successfully deleted the copy right images from the graphical abstract and revised the graphical abstract according to reviewer suggestions in revised manuscript.

Authors successfully added Data availability statement in revised manuscript.

Authors successfully uploaded minimal underlying data set as either Supporting Information files in revised manuscript.

Authors thoroughly copyedit the manuscript for language usage, spelling, and grammar.

In manuscript all the graph made by OriginPro 8 software and standard deviation was also included in graphical representation to make manuscript more scientifically.

All the structure in manuscript made by using ChemDraw Ultra 12.0 software.

Manuscript is now accordance to PLOS ONE style.

Thanks to the editorial board and reviewers for necessary suggestions regarding the manuscript. 

Answer to Reviewer Regarding [PONE-D-20-15280R1]:

In revised manuscript author had cited relevant papers that explain the utility of fruit juices in organic synthesis in introduction part of manuscript and detail of references are given below:

Pramanik T, Pathan AH, Gupta R, Singh J, Singh S. Exploring the utility of fruit juices as green medium for Biginelli reaction. Res. J. Pharm. Bio. Chem. Sci. 2013 5(5):444-449.

Pal R. Fruit juice; A natural, green and biocatalyst system in organic synthesis. Open J. Org. Chem. 2013 1(4);47-56.

Authors also deleted the poor quality references from the manuscript. Quality reference which explain the mechanism of reaction was include in revised manuscript and details reference given as below: 

Sharma N, Sharma UK., Kumar R, Richa, Sinha AK. Green and recyclable glycine nitrate (GlyNO3) ionic liquid triggered multicomponent Biginelli reaction for the efficient synthesis of dihydropyrimidinones. RSC Adv. 2012, 28;1-4.

Authors feel that revised manuscript acceptable for publication and authors declared that there is no conflict of interest regarding publication of this paper.

The revised manuscript is technically sound, scientifically written. Experiments related to biological activity were performed in triplicate and conclusion is well written based on data presented in manuscript.

Statistical analysis (Standard deviation) mentioned in table and graph for making revised manuscript more informative.

All relevant data are within the paper and its supporting information files. Authors made all data underlying the findings in their manuscript fully available.

Authors presented the manuscript in intelligible fashion and written in standard English language.

Financial disclosure statement was also given in cover letter. Authors received no specific funding for this study. The funder has no role in study design, data collection and analysis, decision to publish or preparation of the manuscript. 

All the figures in revised manuscript meet PLOS requirements, according to Preflight Analysis and Conversion Engine (PACE).

Revised manuscript fully meets PLOS ONE’s publication criteria.

Thanks to the editorial board and reviewers for necessary suggestions regarding the manuscript. Please consider all the points mentioned above and if reviewer required any more revision then inform by email.

---

## [Editor Report · Decision Letter 2]

11 Aug 2020

One-Pot Three Component Synthesis of Substituted Dihydropyrimidinones Using Fruit Juices as Biocatalyst and their Biological Studies

PONE-D-20-15280R2

Dear Dr. Gulati,

We’re pleased to inform you that your manuscript has been judged scientifically suitable for publication and will be formally accepted for publication once it meets all outstanding technical requirements.

Kind regards,

A Ganesan

Academic Editor

PLOS ONE